

# Understanding uncertainties in coastal sea level altimetry data: insights from a round robin analysis.

Florence Birol[1], François Bignalet-Cazalet[2], Mathilde Cancet[3,1], Jean-Alexis Daguze[4], Wassim Fkaier[1], Ergane Fouchet[3,5], Fabien Léger[1], Claire Maraldi[2], Fernando Niño[1], Marie-Isabelle Pujol[4], Ngan Tran[4]

5     1 LEGOS, University of Toulouse, IRD, CNES, CNRS, UPS, Toulouse, France
    2 CNES, Toulouse, France
    3 Noveltis, Toulouse, France
    4 Collecte Localisation Satellites (CLS), Toulouse, France
    5 Mercator Ocean International, Toulouse, France

10     *Correspondence to*: Florence Birol (Florence.Birol@univ-tlse3.fr)

**Abstract.** The satellite radar altimetry record of sea level has now surpassed 30 years in length. These observations have greatly improved our knowledge of the open ocean and are now an essential component of many operational marine systems and climate studies. But use of altimetry close to the coast remains a challenge from both a technical and scientific point of view. Here, we take advantage of the recent availability of many new algorithms developed for altimetry sea level computation to analyze the sources of uncertainties of this procedure when approaching the coast. To achieve this objective, we did a round robin analysis of radar altimetry data, testing a total of 21 solutions for waveform retracking, correcting sea surface heights and finally deriving sea level variations. Uncertainties associated with each of the components used to calculate the altimeter sea surface heights are estimated by measuring the dispersion of sea level values obtained using the various algorithms considered in the round robin for this component. We intercompare these uncertainty estimates and analyze how they evolve when we go from the open ocean to the coast. At regional scale, complementary analyses are performed through comparisons to independent tide gauge observations. The results show that tidal corrections and mean sea surface can be significant contributors to sea level data uncertainties in many coastal regions. However, improving quality and robustness of the retracking algorithm used to derive both the range and the sea state bias correction, is today the main factor to bring accurate altimetry sea level data closer to the shore than ever before.

## 1     Introduction

Since the early 1990s, satellite altimetry has routinely observed the ocean surface topography, resulting in a more than 30-year-long record of accurate and nearly-global sea level data. These observations have greatly improved our knowledge of the open ocean and are now a key climate indicator of global warming and an essential component of many operational marine systems (International Altimetry Team, 2021).

30       But in coastal regions, satellite altimetry encounters different technical issues that make it difficult to derive accurate measurements of sea level within tens of kilometers from the land (for example Vignudelli et al., 2011 for a complete review).



Firstly, in the coastal band of a few kilometers wide (corresponding to the altimeter footprint size, i.e. up to about 10 kilometers depending on the satellite altimetry mission), land contamination leads to complex radar waveforms that are difficult to interpret in terms of geophysical parameters through the common process called retracking (Deng and Featherstone, 2006;

Gommenginger et al., 2011). The other main limitation is related to the geophysical and environmental corrections that need to be applied to the altimeter measurements to compute the height of the ocean surface (e.g. wet troposphere, ionosphere, sea state bias, inverse barometer, high frequency wind effect and tides) and that often become inaccurate close to the coast (e.g. Vignudelli et al., 2005; Andersen and Scharroo, 2011). Finally, the traditional use of sea level anomalies (SLA) in oceanography applications requires the removal of a time-average of the height of the ocean surface, called the mean sea

surface height (MSSH). In near-shore areas, the MSSH is contaminated by the same suite of retracking and correction errors as those that arise in the process of computing the height of the ocean surface (Andersen and Scharroo, 2011; Gómez-Enri et al., 2019).

Filling the altimetry data gap in the coastal zone is needed to explain, estimate, and plan for coastal impacts associated with sea level changes induced by ongoing global warming and has motivated a number of coastal altimetry studies, inducing

significant progress during the last decade (Cipollini et al., 2017; Birol et al., 2021). In order to address the limitations mentioned above, new retracking algorithms have been developed to reduce the contamination of spurious signal components in the coastal zone (Passaro et al., 2014; Peng and Deng, 2018; Thibaut et al., 2021). In parallel, significant improvements have also been achieved in altimeter corrections (e.g. wet troposphere and ocean tide corrections, sea state bias), allowing to obtain more accurate altimetry-derived coastal sea level data (Fernandes et al., 2015; Carrere et al., 2016; Passaro et al., 2018).

New MSSH products are also available (Sandwell et al., 2017; Schaeffer et al., 2023). These efforts improve the performance of altimetry in the coastal ocean with respect to the standard solutions provided by space agencies and operational altimetry services. Some of the new algorithms developed are progressively introduced in the operational processing baselines. However, the metrics used to measure the performance improvement and the coastal area, generally change from one study to the other, making it difficult to provide an objective comparison of their relative merits.

Today, several algorithms are available for calculating the range, for most of the geophysical corrections and for the MSSH used to derive coastal SLA from altimetry measurements. The main objective of this paper is to take advantage of them to better understand the sources of uncertainties in the sea level computation when approaching the coast. Particular attention is also paid to the transition between the open ocean and coastal ocean. To this end, a round robin exercise has been done for the components of the altimetric SLA for which several solutions existed. In each case, as many algorithms as possible were

tested with similar metrics to have a common analysis methodology. For each component, we can then objectively discuss the relative performance of the different algorithms in terms of the selected diagnostics. But, assuming that the differences between algorithms reflect the associated uncertainties, we can also analyse and compare how these uncertainties are then reflected in the calculation of the SLA data as we get closer to the coast.





This paper is organized as follows: The objectives and the data used in the round robin analysis are described in Section 2. The methodology is presented in Section 3. Results and discussions for each of the sea level components evaluated are provided in Section 4. Section 5 summarizes the main results and gives some perspectives.

## 2        Objectives and input data of the round robin analysis

### 2.1       General goals

Altimetry technologies have considerably evolved in the recent years with the Delay-Doppler Mode (or SAR for Synthetic Aperture Radar) and the SAR Interferometric Mode (SARIn). Here, we have chosen to focus on the conventional Low Resolution Mode (LRM) technique only, because it has the largest time span and number of altimetry missions, with seamless continuity from the first generation of climate reference altimeters (in the 1990s) until today. For this reason, it also provides the largest number of algorithms available to derive geophysical parameters from corresponding altimetry measurements, including some specific developments to improve the data quality near the coast (Fernandes et al., 2015; Passaro et al., 2014).

The Round Robin exercise presented in this study was implemented to inter-compare algorithms used to calculate the SLA from LRM altimetry measurements in order to evaluate their accuracy. In what follows, we will not go into the technical details of the radar altimetry techniques, as it is thoroughly explained elsewhere (e.g. Fu and Cazenave, 2001). In summary, satellite altimetry is based on a radar altimeter sending/receiving pulses/echoes towards/from the overflown surface. By analyzing the backscattered echoes (the so-called waveform), we can deduce the altimeter range (i.e. the distance between the satellite's center of mass and the mean reflected surface) through a process called retracking. The transformation from the range into SLA then requires knowledge of auxiliary information (e.g., satellite altitude; atmospheric and geophysical corrections; MSSH). Finally, the SLA is computed according Eq. (1):

*SLA= Altitude of satellite – Altimeter range – Ionospheric correction – Dry tropospheric correction – Wet tropospheric correction – Sea state bias correction – Solid earth tide correction – Geocentric ocean tide correction - Geocentric pole tide correction – Dynamic atmospheric correction – Mean sea surface height*        (1)

Each of the terms of Eq. 1 will be called "SLA component" hereinafter. Any systematic error in each of these terms directly results into errors in the SLA estimates. The SLA components are all derived from numerical or empirical models, or from altimetry or auxiliary observations. For most of them, different solutions exist. When available, we have included solutions developed specifically for the coastal environment (see section 2.2). The algorithms are then inter-compared with common metrics (see section 2.3).

This study focuses on altimetry data on the coastal zone, whose definition varies widely from one study to the other (Laignel et al., 2022). Here, to take a broad reference, we define the global coastal zone as the geographical area between the coastline and 200 km offshore, at global scale (Figure 1). Because they provide SLA data closer to the coast (*Birol et al., 2021*), we consider along-track altimetry measurements at the original high-frequency sampling rate (20Hz). They are the raw



data from which the widely used 1Hz datasets are obtained through averaging, so the results of the present study should remain
      valid for those datasets.

      Coastal conditions being different from one region to the other, the uncertainty sources in altimetry sea level may have
      a marked geographical dependency. It was consequently decided to carry out this study at both global and regional levels. For
      this purpose, three coastal areas were chosen because of their very different coastal and oceanographic contexts, and because
of the availability of regional ocean tide corrections (section 2.2): the Mediterranean Sea, the North East Atlantic Ocean and
      Eastern Australia (Figure 1).

      Moreover, in order to estimate the degree of agreement of our results from one altimeter to another, we consider data
      from the two reference missions Jason-2 and Jason-3 as they flew on the same nominal orbit (see below).

      Finally, note that the focus of this round robin study is the comparison of different processing solutions in order to gain
insight about the sources of uncertainties in sea level data when approaching the shoreline. Even though we compare the
      algorithms against each other with a set of performance assessment criteria, we do not aim to identify here the "best" algorithm
      among all those tested to compute SLA. The ranking of algorithms depends mainly on user needs, which in turn define the
      best set of metrics to use in that particular case. The final choice for a given application can then only be a trade-off between
      different criteria (e.g. computational cost, availability of the algorithm/solution, continuity between altimetry missions,
improvement of long scales over white noise, …).

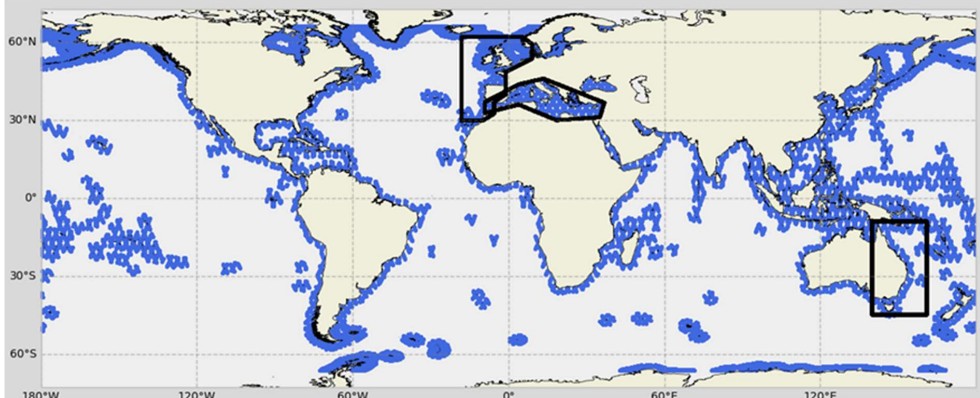

**Figure 1:** Geographical domains covered by the Round Robin study. The North East Atlantic, Eastern Australia and Mediterranean Sea used in the regional analyses are indicated by black squares. All of them comprise the [0-200 km] coastal band except the Mediterranean Sea which is complete, Black Sea excluded.




## 2.2  Overview of the selected algorithms

To achieve the objectives of this study, it was crucial to have access to as many algorithms as possible, including those used in the operational sea level products (i.e. the level 2 Geophysical Data Records or GDR, see https://www.aviso.altimetry.fr). To collate all the data, we used the CNES (French Space Agency) internal altimetry database that contains all the operational Jason-2 and Jason-3 GDR products, where we could add project-oriented datasets that were made available for the purpose of this study (e.g. outputs of the ALES and Adaptive retrackers, regional tide solutions; see Table 1 for complete information). For an objective evaluation of the results from the metrics computed for both Jason-2 and Jason-3, we have selected 3 years of data (i.e. 111 cycles) for each of these altimetry missions. For Jason-2, cycle 193 (start: 27/09/2013) to cycle 303 (end: 02/12/2016) have been chosen. For Jason-3, the dataset covers cycle 1 (start: 17/02/2016) to cycle 111 (end: 22/02/2019).

Concerning the SLA components of Eq. 1, the Altitude of satellite, Dry Tropospheric Correction and the Dynamic Atmospheric Correction were not be included in the round robin because only one solution was available for each of them. The Solid earth tide height and the Geocentric pole tide height were also discarded because considered as very accurate and non-critical for coastal sea level calculations (Andersen and Scharroo, 2011). For the other components, the main criterion to select the algorithms was the availability of the corresponding dataset at global scale and for the whole study time period (i.e. 27/09/2013 to 22/02/2019). A few exceptions have been made for specific reasons explained below.

The Altimeter range and the Sea State bias correction (SSB) derived from the ALES retracker (Passaro et al., 2014) are part of the ESA CCI Coastal Sea Level product (Cazenave et al., 2022). These datasets are not global but cover a large part of the coastal ocean (except latitudes above 60°N, Japan, Alaska and the Okhotsk and Bering Seas zones in the north, and New Zealand, Antarctica and some small islands in the south, as shown in Figure 1 of the aforementioned article). Because the ALES retracker has been shown to improve coastal altimetry sea level retrieval in comparison with the standard MLE4 (Maximum Likelihood Estimator) retracking algorithms (Passaro et al., 2015), this study would not be complete without its inclusion. As a consequence, all the algorithms concerning the altimeter range and the SSB will be evaluated only where ALES data are available.

The Geocentric ocean tide is one of the main contributors to the sea level variations in regions with strong tidal motions, which is the case of a large part of the global ocean continental shelves. Tides must be removed from satellite altimetry data using hydrodynamic models in order to avoid aliasing issues with other ocean dynamics signals (Chelton et al., 2001). Even if tidal modelling benefited from many improvements these last years, the most recent global models still show errors of several centimetres in coastal regions (Stammer et al, 2014). The development of regional models at higher resolution improves the estimation of ocean tides on the continental shelves and consequently provides more accurate altimetry corrections (Cancet et al., 2018). Including such regional tidal models in this study allows us to analyze the uncertainties associated with the use of global tidal models in computing altimetry coastal sea level. The ocean tide correction from regional tidal model is, by definition, available only in its geographical area. For this project, regional tidal corrections were made available by CNES/Noveltis for the Mediterranean Sea, the North East Atlantic and Eastern Australia regions. The evaluation of all the



algorithms concerning the ocean tide correction will then be done only at regional scale. For comparisons at global scale,
150    readers can for example refer to Stammer et al, 2014 and Lyard et al., 2021.

Concerning the SSB, some of the most recent datasets (i.e. MLE4 2D 20Hz, MLE4 3D 20Hz, Adaptive 3D 20Hz)
currently exist as prototypes only for Jason-3 and not for Jason-2. Given that the SSB is identified as a large source of
uncertainty in altimetry sea level retrieval, particularly 10-15 km from the coast (Andersen and Scharroo, 2011; Passaro et al.,
2018), it was decided to include these algorithms in this study. As a consequence, the metrics concerning the SSB will be
155    computed only for Jason-3.

Finally, the SLA components and algorithms used in this round robin are listed in Table 1. They represent a total of 6
components and 21 algorithms.

| SLA Components | List of algorithms tested |
|---|---|
| **Altimeter Range** | **3 solutions:**<br>• MLE4 – in GDR product<br>• Adaptive (*Tourain et al., 2021*) – in GDR product<br>• ALES (*Passaro et al., 2014*) – version ESA CCI Coastal Sea level product |
| **Ionospheric correction** | **2 solutions:**<br>• Dual-frequency, filtered\* – in GDR product<br>• GIM (*Ijima et al., 1999*)\* – in GDR product |
| **Wet tropospheric correction** | **3 solutions:**<br>• Radiometer\* – in GDR product<br>• 3D ECMWF model\* – in GDR product<br>• GPD+\* (*Fernandes et al, 2015*) – from AVISO+ 2022 |
| **Ocean tide correction** | **4 solutions:**<br>• EOT20 (*Hart-Davis et al., 2021*)<br>• FES2014b (*Lyard et al., 2021*) – in GDR<br>• FES2014b, unstructured mesh version (*Lyard et al., 2021*), provided by Noveltis<br>• CNES/Noveltis regional models (NEA, Mediterranean Sea, Australia), provided by Noveltis |
| **Sea State Bias (SSB) correction** | **6 solutions:**<br>• MLE4 2D 1Hz\* - in GDR product<br>• MLE4 2D 20Hz (*Tran et al., 2019*), provided by CNES<br>• MLE4 3D 20Hz, provided by CNES<br>• Adaptive 2D 20Hz (*Thibaut et al., 2021*), provided by CNES<br>• Adaptive 3D 20Hz, provided by CNES<br>• ALES 20Hz (*Passaro et al., 2018*) – version ESA CCI Coastal Sea level product |
| **Mean Sea Surface Height (MSSH)** | **3 solutions:**<br>• CNES_CLS15\* (Pujol et al, 2018) – in GDR product<br>• SIO\* (Sandwell et al, 2017)<br>• CNES_CLS22\* (Schaeffer et al., 2023) – provided by CNES |

**Table 1:** SLA components included in the Round Robin exercise (column 1), with the list of algorithms tested for each one (column 2). The
reference algorithms currently used in operational sea level products for each component are underlined. The fields marked with an asterisk
160    (\*) were provided at 1Hz only and have been linearly interpolated to 20 Hz for the purposes of this study; the others were at 20Hz. GDR is
the official Geophysical Data Record product distributed by the space agencies (version D for Jason-2 and version F for Jason-3).



### 2.3    Tide gauge data

At regional scale, some of the metrics used to estimate the accuracy of altimetry coastal sea level derived with the different algorithms listed in Table 1 are based on comparisons with independent hourly sea level observations from tide gauges. Tide gauge measurements from the following databases have been used:

- Mediterranean Sea: CMEMS (https://marine.copernicus.eu), Refmar (http://refmar.shom.fr/en/home) and ISPRA (https://www.mareografico.it/);

- North East Atlantic Ocean: BODC (https://www.bodc.ac.uk/), Refmar (http://refmar.shom.fr/en/home) and UHSLC (https://uhslc.soest.hawaii.edu/);

- Eastern Australia region: UHSLC (https://uhslc.soest.hawaii.edu/) and BOM (http://www.bom.gov.au/).

For all these databases, the tide gauge stations selected for this study had to meet the following selection criteria (all of them must apply):

- Quality data available over the whole study period (2013-2019); time series with many data gaps longer than 5 days were not considered.

- Stations located at a distance shorter than 50 km from a Jason2/3 nominal track and not too deep inside estuaries or sheltered by islands, so that the ocean dynamics signals captured by the in situ instrument and the satellite altimeter are as similar as possible.

From all the considered databases, 13 stations met these criteria in the North East Atlantic region, 12 in the Mediterranean Sea and 8 in the Eastern Australia region (see Figure 2).

To compare the altimetry and tide gauge sea level measurements, the tidal signal has been removed from the tide gauge sea level time series using a harmonic analysis approach. The effect of atmospheric pressure and wind on the tide gauge sea level has been removed using the same correction as for the altimetry observations (Dynamic Atmospheric Correction from MOG2D solution, LEGOS/CNRS/CLS, 1992; Carrère and Lyard, 2003).



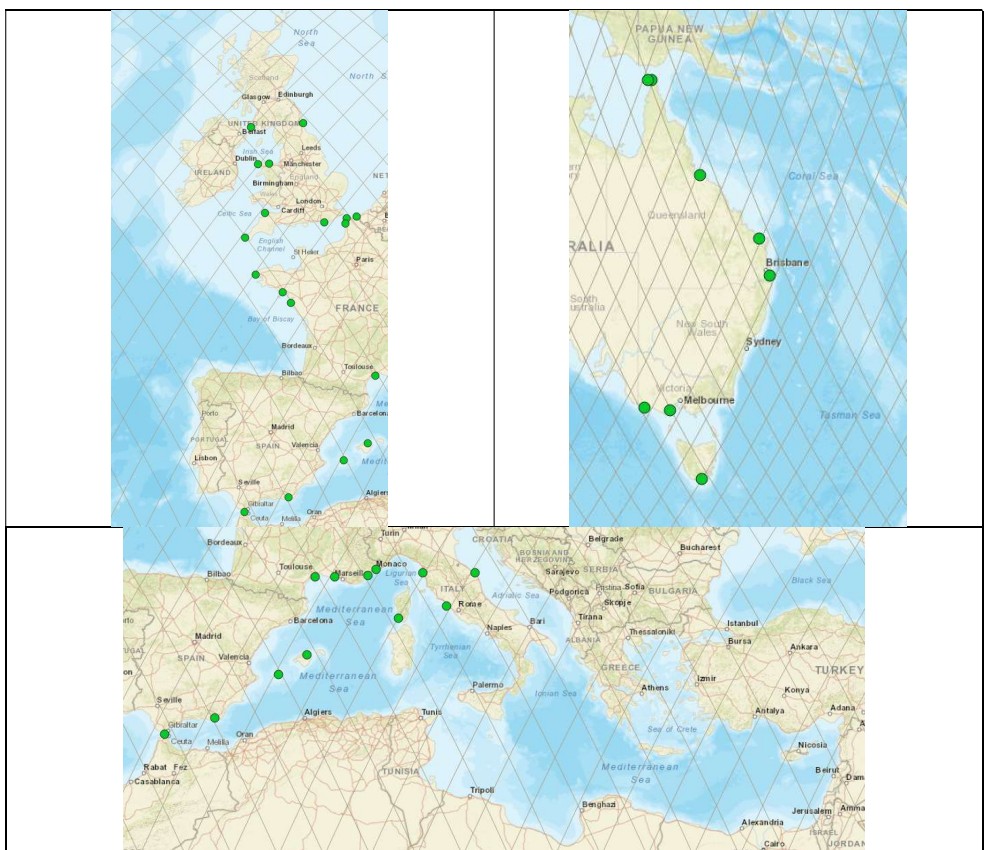

**Figure 2:** Jason-2/3 nominal tracks (black dashed lines) and tide gauge stations (green dots) used in this round robin study in the North East
Atlantic (top left), Eastern Australia (top right) and Mediterranean Sea (bottom).

## 3        Methodology

The basic principle of this round robin study is to compare all the selected SLA components and algorithms using the same
metrics, so their impact on the coastal sea level computation can be assessed in the same way. In order to measure the
consistency of all the results between different altimetry missions, the same analysis has been done for both Jason-2 and Jason-
3 (with exception of the SSB component, see section 2.2), at global and regional scales (i.e. in the three regional domains
shown in Figures 1 and 2). For each SLA component, the accuracy is investigated by analyzing the dispersion of SLA values



we obtain using the various corresponding algorithms mentioned in Table 1, with focus on the coastal ocean. At regional scale, the analysis is completed with a comparison to independent tide gauge observations.

In practice, the study has been organized by SLA component. At global scale, for each of them, the different algorithms have been first inter-compared in terms of data availability (spatial pattern of the data availability, data availability as a function of distance to the coast) and general statistics (mean, standard deviation, histograms of values). Then, the impact on the SLA calculation has been analyzed for each algorithm tested for this component. Therefore, only one term (algorithm) of the SLA definition (Eq. 1) changes at a time. All the other SLA components are the state of the art of the operational sea level products at the time this study was conducted (see algorithms that are underlined in Table 1). They are considered here as the reference algorithms. At regional scale, the inter-comparison between the different algorithms has been not only done in terms of data availability and general statistics, but also in terms of comparison to the tide gauge measurements (statistics and local altimetry data availability).

Before carrying out statistical analyses, because original altimetry measurements are not sampled exactly at the same points at each cycle, all the along-track sea level component and SLA values were binned along average ground tracks of the Jason missions with a resolution of 20Hz (i.e. ~0.3 km). When computing metrics on the SLA components, no editing was applied and all values available in the dataset were used. For the metrics on the SLA itself, values outside the window [-3m ; 3m] were systematically discarded everywhere. In the Mediterranean Sea, associated with generally lower SLA variations, a stricter window [-1m ; 1m] was applied. For each SLA point time series, outliers outside a 4 σ window have also been removed from the computations, σ  being the standard deviation of the SLA time series. Finally, altimetry points have been binned considering their distance to the coast (Figures 3, 5, 6, 9, 10, 11). To ensure robust global or regional statistics, we considered a fixed number of altimetry points in each bin, with the bin size varying from about 300 m at the coast to 1.2 km at 200 km from the coast, as the density of altimetry points is higher close to the coasts due to the presence of islands and to the tracks configuration.

Concerning the comparison between altimetry and in situ SLA, for each tide gauge station, the nearest satellite track to the station is selected. Only altimetry data located at a distance to the coast shorter than 20 km and at a distance to the tide gauge station shorter than 40 km are used.

In the end, we have evaluated 21 algorithms at global scale and for the three study regions, for both Jason-2 and Jason-3 missions (when possible). The total number of inter-comparison diagnostics reaches several hundred. As it represents a considerable amount of work that can be useful for purposes other than those of our study, they all have been made available on https://www.aviso.altimetry.fr/en/data/products/sea-surface-height-products/global/altimetry-innovative-coastal-approach-product-alticap/roundrobin-reports.html, so that colleagues can use them for other applications. In the following section, we will only show some of the results obtained in line with the objectives of this study.



## 4        Results

### 4.1     Ionospheric correction

The commonly used method to compute and correct the altimeter range for the delay effects due to the ionosphere is the linear combination of data measured at two different radar frequencies (Chelton et al., 2001). The corresponding dual-frequency correction is considered less accurate in coastal areas due to altimeter echoes and a required along-track filtering (Fernandes et al., 2014), and both can be altered by land contamination in these areas. A second method consists of using external models, and the GNSS-based Global Ionospheric Maps (GIM; Komjathy & Born, 1999) are the most commonly used corrections for single-frequency altimeters and for coastal and inland applications (Dettmering and Schwatke, 2022). In the following, we estimate the relative uncertainties of these two solutions for the ionospheric correction as they approach the coast by comparing their respective statistics.

Here, only the example of Jason-2 is presented. Figure 3 shows the global mean of the standard deviation (STD) of the SLA computed with each of the two corrections (top and middle panels a and b) and the spread of the differences of these STD of SLA (bottom panel c), as a function of the distance from the coast. Note that the increase in the STD values below 10 km from the coast (Figure 3b, middle) is generally observed in this type of diagnostic and is largely related to an increase in the SLA errors near the coastlines (Andersen and Scharroo, 2011). It integrates the errors of all the different SLA components and does not necessarily reflects firstly those of the ionospheric correction. However, between the 2 solutions, we observe a difference of 0.1 cm in STD beyond 20 km from the coast, which then increases up to ~0.75 cm in the last 5 km.

The standard deviation of the differences obtained between the SLA solutions (Figure 3.c) also clearly increases when approaching the coast. The corresponding spread values remain below 0.2 cm in the open ocean up to 40 km from the coast, then range between 0.2 and 0.7 cm between 10 and 40 km, and finally increase up to 2.8 cm in the last 10 km. These numbers can be considered as an estimate of the SLA uncertainty due to the ionospheric correction.



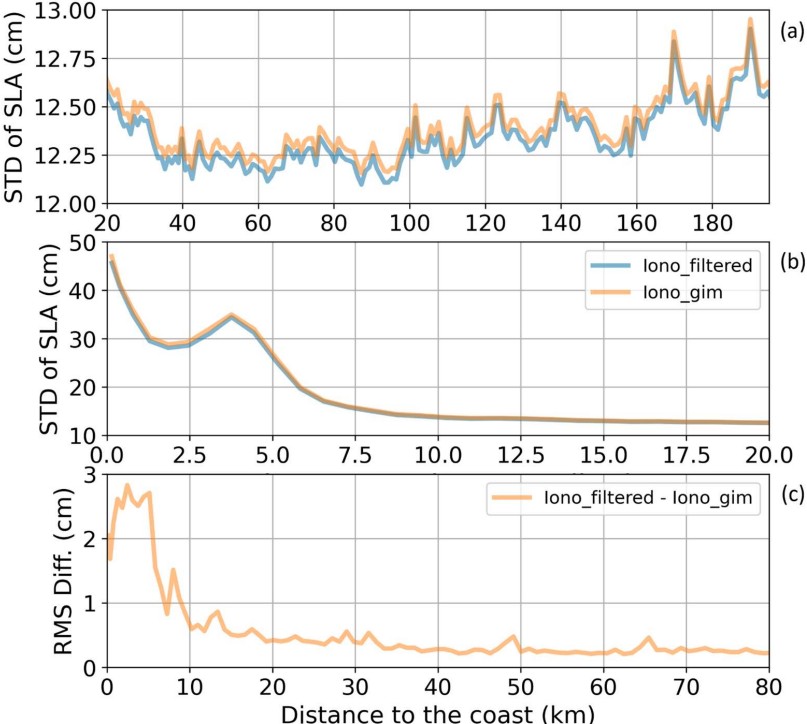

**Figure 3:** (a) and (b): Global mean of standard deviation values (in cm) of the SLA along all Jason-2 tracks for the period 27/09/2013 to 02/12/2016 when applying different ionospheric corrections (dual-frequency filtered in blue, GIM in green). Results are represented as a function of the distance to the coast (in km) between 200 km and 20 km from the coast (a) and between 20 km and 0 km from the coast (b). (c): Global standard deviation of the differences of standard deviation values (in cm) of the SLA when applying the different ionospheric corrections between 80 km and 0 km from the coast.

Figure 4 illustrates that this average result at global scale presents some geographical features, with lower/larger SLA STD values generally obtained with the dual-frequency solution below/over 15-20°N/S. These latitudinal patterns are very consistent with the large-scale features of the mean and variability of the ionospheric corrections (Fernandes et al., 2014). When analysing the tide gauge comparison statistics, no significant differences were found between the two SLA datasets (not shown here, but see the reports under the link provided at the end of section 3). This could be due to the fact that the regions considered in our study show small differences between the ionospheric solutions (see NEA, Mediterranean Sea and Eastern Australia on Figure 4), as are potentially the associated uncertainties.





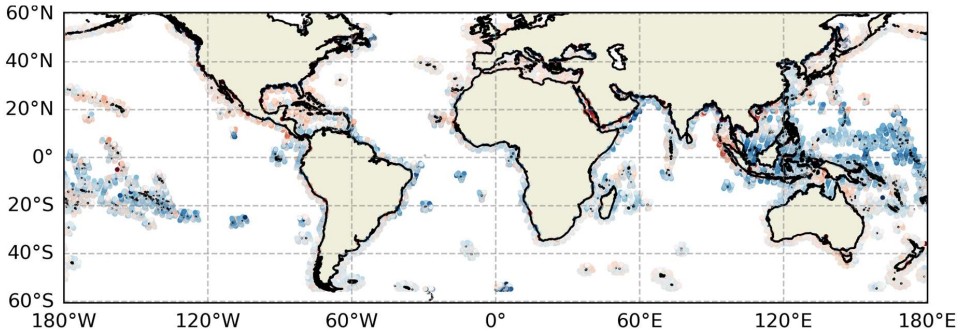

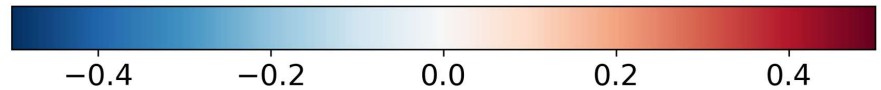


**Figure 4:** Map of the differences in standard deviation of the SLA (in cm) along all Jason-2 tracks for the period 27/09/2013 to 02/12/2016 when applying the dual-frequency ionospheric correction compared to when using the GIM model. Results are only shown between between 200 km and 0 km from the coast.

**4.2    Wet tropospheric correction**

The wet tropospheric correction (WTC) is related to the path delay in the altimeter return signal due to cloud liquid water and water vapour in the atmosphere. It can be derived either from meteorological models or from a microwave radiometer onboard the altimetry mission. Due to the large space-time variability of this correction (0-50 cm), the latter is considered the best option. Lazaro et al., 2020 report an associated reduction of 1.2-2.2 cm2 in the SLA variance on average between 0 and 200

km from the coast. However, because of the radiometer footprint, this WTC is known to be decrease in quality starting at ~50 km from the coast, leading to errors of several cm in the SLA (Andersen and Scharroo, 2011, Obligis et al., 2011). The importance of coastal zones has recently motivated the development of dedicated strategies to solve the WTC issue in land/sea transition areas (Obligis et al., 2011; Cipollini et al., 2017; Maiwald et al., 2020). One approach consists in combining data from several sources through objective analysis to estimate the WTC where it is invalid or not defined. The most mature global

dataset based on this approach and available for many altimetry missions is the so-called GPD+ (GNSS derived Path Delay) product (Fernandes et al, 2015). Here, we compare the metrics obtained with three WTC solutions: the radiometer-derived correction, the correction computed from the ECMWF model and the GPD+ correction. Again, only the example of Jason-2 is presented since the numbers obtained with Jason-3 are globally the same.

In Figure 5 (a and b), representing the global mean of the STD of SLA associated with the three WTC corrections as a

function of the distance from the coast, we observe that the differences between the three solutions are ~0.1 cm up to the coast.





They can reach several centimeters very locally (not shown). The GPD+ and radiometer solutions are very close and allow to reduce the STD of SLA in comparison to ECMWF, which confirms results of Lazazo et al (2020). Here again we use the spread of the differences of STD of SLAs for each couple of SLA solutions as a proxy of the SLA uncertainty associated with the wet tropospheric correction (Figure 5.c). In general, the results are very stable between 200 km and about 7.5 km from the

coast, with values below 0.3-0.5 cm. A clear increase occurs in the last 7.5 km, with maximum values reaching 1.7 cm. The GPD+ and radiometer solutions show the best agreement from 200 to 12 km to the coast, with spread values of around 0.2 cm. While the spread between GPD+ and ECMWF solutions remains relatively constant, the radiometer solution starts to disagree with the two others at about 7.5 km from the coast, with maximum values of spread reaching 1.7 cm. As for the ionospheric correction, no significant differences were found in tide gauge comparison statistics.


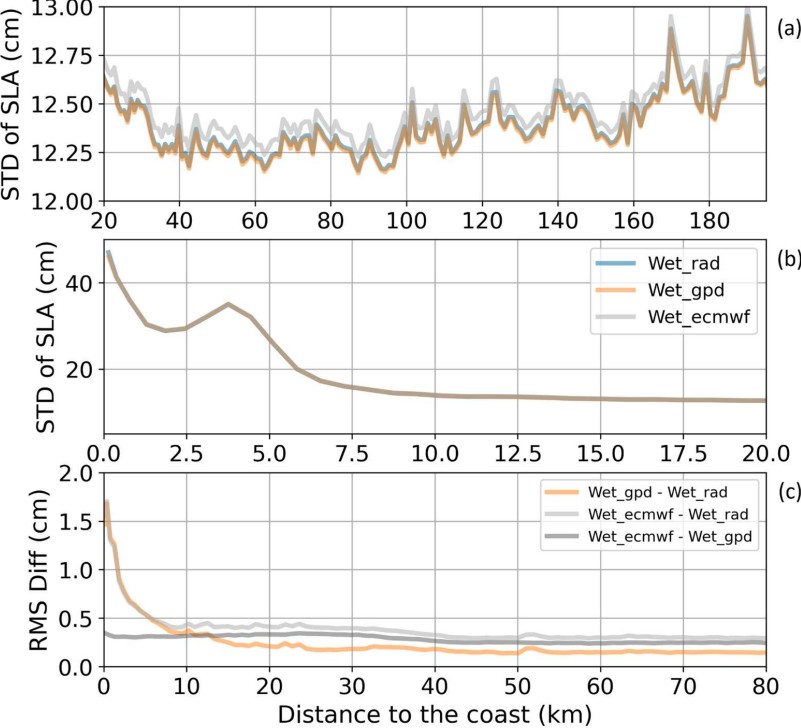

**Figure 5:** (a) and (b): Global mean of standard deviation values (in cm) of the SLA along all Jason-2 tracks for the period 27/09/2013 to 02/12/2016 when applying different WTC corrections. Results are represented as a function of the distance to the coast (in km) between 200 km and 20 km (a) and between 20 km and 0 km from the coast (b). (c): Global standard deviation of the differences of standard deviation

values (in cm) of the SLA when applying different WTC corrections between 80 km and 0 km from the coast.



As a conclusion of this section, Figure 5 illustrates the progress that has been made on the WTC quality in nearshore regions. In 2011, Andersen and Scharroo reported a deterioration of half its quality at 30 km from the coast. Our results show that today, WTC uncertainties increase only 10 km from the coast (on average). Note however that this conclusion must be modulated by one consideration. The results associated with the radiometer solution might be different for altimetry missions

which have not been reprocessed recently with the GDR product versions used here (cf Table 1).

### 4.3    Ocean tide correction

Tides must be removed from altimetry measurements of sea level to avoid aliasing effects, as the satellite sampling period (9.9 days at best) does not allow to resolve them. For this purpose, solutions from several global models can be used (a comprehensive summary can be found in Stammer et al., 2014, and in Zaron and Elipot, 2020 for more recent models). Global

tidal models have largely evolved since the early days of altimetry, and today they all reproduce open ocean tides with an accuracy of approximately 1-2 cm (Andersen and Scharroo, 2011). However, in shallow waters, they show larger differences (Ray et al., 2011) and may have errors larger than 10-20 cm (Ray, 2008), due to poorly resolved bathymetry and more complex tidal hydrodynamic features that are difficult to model. Different studies have shown that regional tidal models generally show better performances in coastal areas, compared to global models (Cancet et al., 2018; 2022).

Here, in order to investigate this correction in coastal areas, we inter-compare two global tidal models (EOT20, Hart-Davis et al., 2021 and FES2014b, Lyard et al., 2021) and a CNES/Noveltis regional solution for the Mediterranean Sea, NEA and Eastern Australia (Cancet et al., 2022). Because the resolution of the tidal model grid can have an impact on the tidal estimates in coastal regions, where the tidal spatial features are smaller, two versions of the global FES2014b model have been considered: 1) interpolated on a regular 1/16° grid (i.e. about 7.5 km) as it is officially distributed and used in the operational

altimetry products, and 2) on the native unstructured grid, whose resolution spans between ~4 km and ~15 km in coastal regions. This part of the study has been restricted to the three regions where the regional solutions are available. Here we mainly present results on the NEA region which is one of the coastal zones where the tides are the strongest and one of the most difficult to model in the world. Hence the results are more contrasted than for the two other regions. We only show results for Jason-2; all the results for the three regions and the two missions are available on line

(https://www.aviso.altimetry.fr/en/data/products/sea-surface-height-products/global/altimetry-innovative-coastal-approach-product-alticap/roundrobin-reports.html).

Figure 6 (a and b) shows the STD of SLA when applying each of the 4 tidal corrections in the NEA region. EOT20 is systematically about 0.5 to 2.5 cm above all the other solutions; (the maximum of 2.5 cm is reached at 6-7 km from the coast, relative to the regional solution). In the open ocean, the difference between the three other solutions is below 0.5 cm and this

value increases towards the coast, reaching 1 cm at 3 km from the coast. The systematic difference between EOT20 and the other models may be at least partly due to its tidal spectrum which is smaller (17 tidal components available, 15 used for this study for reasons of incompatibility with the Dynamic Atmospheric Correction) than that of the FES2014b and regional models (all with 34 tidal components), thus removing less tidal signal from the altimetry SLA data. Indeed, the omitted tidal




components in EOT20 are secondary and non linear elements that generally present larger (at the millimeter or centimeter
level) amplitudes in shallow waters than in the deep open ocean (sub-millimeter).

The spread between the different SLA solutions obtained with these four tidal corrections is spatially variable, ranging
between 1 and 2 cm up to 20 km from the coast (Figure 6.c). Below this distance the spread increases, with values reaching
about 4 cm when considering only tidal solutions with the same spectrum, and 5 cm when considering also the EOT20 model.

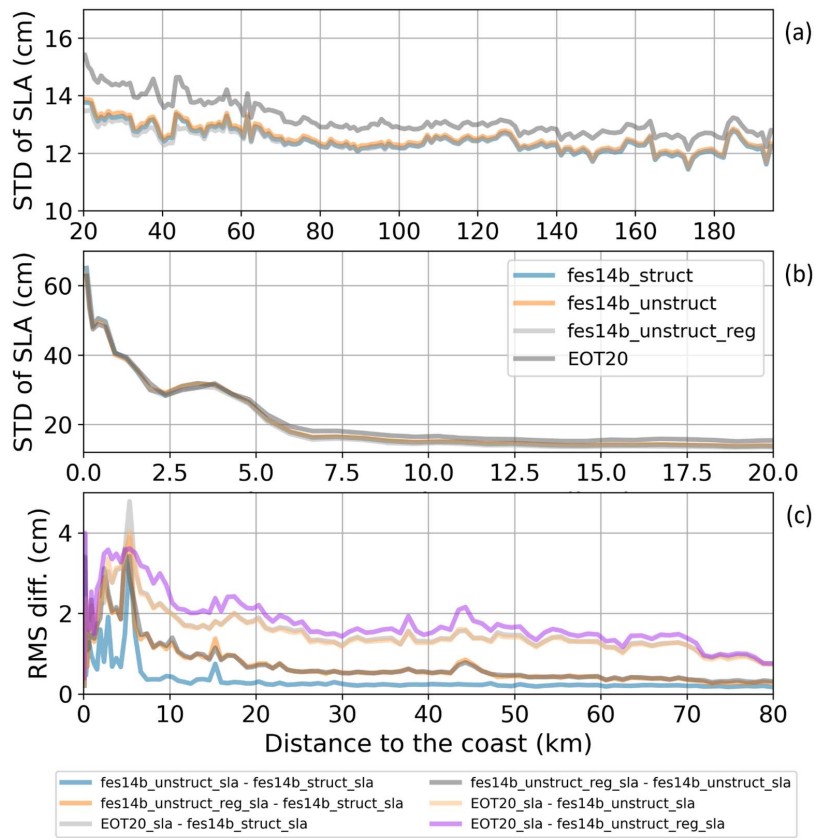


**Figure 6:** (a) and (b): Regional mean of standard deviation values (in cm) of the SLA along all Jason-2 tracks for the NEA region and for
the period 27/09/2013 to 02/12/2016 when applying different ocean tide corrections. Results are represented as a function of the distance to
the coast (in km) between 200 km and 20 km from the coast (a) and between 20 km and 0 km from the coast (b). (c): Regional standard deviation of the
differences of standard deviation values (in cm) of the SLA in the NEA region when applying different tide corrections between 80 km and
0 km from the coast.

Figure 7 represents the regional structure of the differences in the STD of SLA corrected with the regional tidal solution and with the FES2014b global solution on the native unstructured grid (regional model minus global model). In reddish (blueish) regions, FES2014b (the regional solution) decreases the STD of SLA more significantly. The differences are in the
order of a few millimeters in most parts of the NEA region, except in shallow areas where the tidal amplitudes are the largest (English Channel, Celtic Sea, southern part of the North Sea). In these regions, the differences provide negative values that vary significantly in space, from ~1 cm to more than 3 cm.

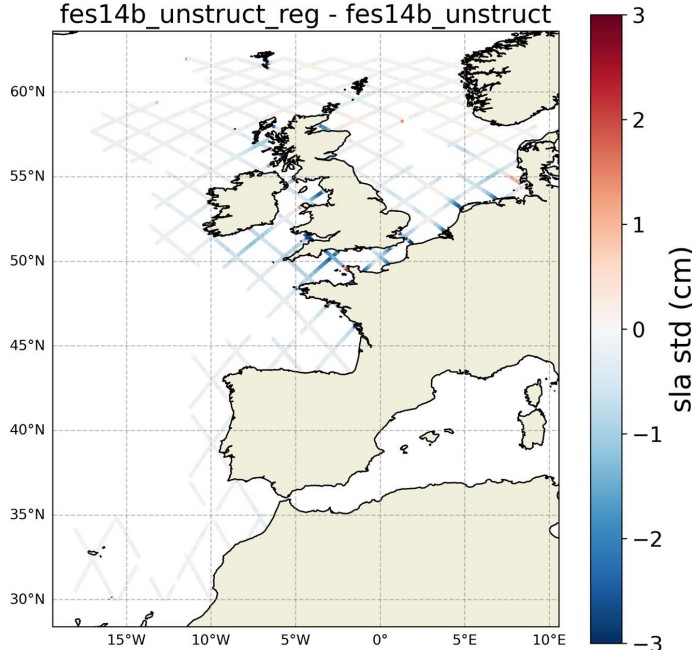

**Figure 7:** Regional map (NEA coastal area) of the differences of standard deviation values (in cm) of the SLA along all Jason-2 tracks for
the period 27/09/2013 to 02/12/2016 when applying the CNES/Noveltis regional model compared to applying the FES2014b global tidal model on its native unstructured grid (regional model minus global model).

These results illustrate that if very significant progress has been made since studies as Ray (2008), large uncertainties remain in ocean tidal corrections in coastal regions, linked to the model accuracy, but also for other reasons such as the tidal
spectrum used. These uncertainties have complex spatial structures, associated with the tidal signal itself, which makes it meaningless to estimate them on a global scale. They are local in nature and can be of a few mm in the Mediterranean Sea, a few cm in the Tasman Sea and on the Northeastern Australian shelf (not shown) or even larger (English Channel).



To further quantify these geographical disparities, the 4 SLA solutions were compared with the equivalent series of sea
level variations from tide gauges in the three study regions (see section 2.3 and Figure 2 for the details on the tide gauges
selection). Here again, the comparison is made in terms of statistics: correlation and root mean square (RMS) differences
between the altimetry and in-situ SLA observations (Figure 8). The statistics are calculated for each 20Hz altimetry point
corresponding to the selection criteria specified in section 2.3 and then averaged by tide gauge and by region. The results show
that the choice of the tidal model in the SLA calculation has more impact in the NEA than in the two other regions. However,
if this result can probably be extrapolated to the whole Mediterranean Sea, which is characterized by small tidal amplitudes
except in a few areas (Adriatic Sea, Gulf of Gabes), it should be qualified for the Australia region, as it strongly depends on
the tide gauge stations. Indeed, for our study we could select 8 stations close to the Jason2/3 nominal ground tracks that
happened to be located in regions with rather low tidal signatures. That may thus not be representative for other Jason-2/3
ground tracks (or even other missions) that may sample Australian regions with larger tidal signals and bigger uncertainties in
the models.

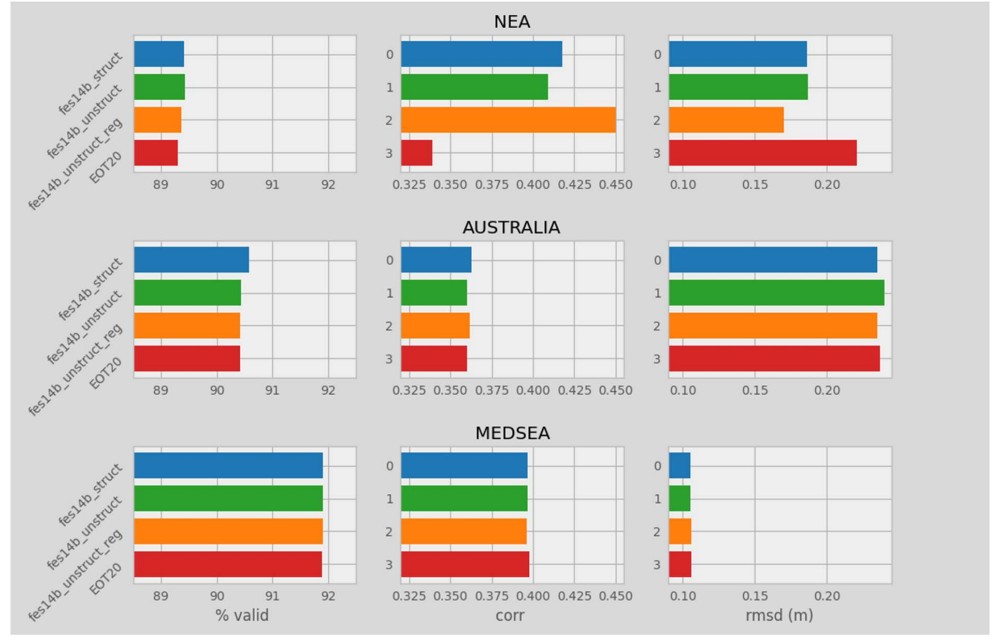


**Figure 8:** Regional averages of left) the % of altimetry SLA data available in the time series for all the coastal zones selected, middle)
correlation values and right) RMS of the differences between the altimetry and in-situ SLA observations.



### 4.4    Mean Sea Surface Height

MSSH models correspond to the relative steady-state sea level and are obtained by time averaging and interpolating the instantaneous sea surface height data observed by the different altimeters over a finite period (Andersen and Knudsen, 2009). The precision and grid size of the existing MSSH solutions have been gradually improved and enhanced with the development of satellite altimetry. For wavelengths shorter than 250 km, their error is of the order of 1–2 cm2 (Pujol et al., 2018) but it can become larger near the coasts where MSSH solutions suffer from the decrease in the quality and quantity of SLA data used to

calculate them.

The investigation of the impact of coastal MSSH errors in the corresponding altimetry SLA data lies here on comparing 3 models: CNES_CLS15 (Pujol et al., 2016), SIO (Sandwell et al., 2017) and CNES_CLS22 model (Schaeffer et al., 2022). Figure 9 (a and b) shows the global average of the STD of SLA as a function of the distance to the coast, applying each of the three MSSH solutions. The three plots are almost identical. Concerning the spread between the SLA solutions, it is below 0.5

cm in the open ocean (Figure 9.c). It starts to increase between 20 and 8 km from the coast, with values between 0.5 and 1 cm, and then rapidly amplifies in the last 8 km to the coast, with values reaching about 4 cm. It highlights that discrepancies between the MSSH solutions are concentrated in the coastal regions. We note that, close to the coast, the spread between the two CNES-CLS MSSH solutions is lower (2 cm) than the spread with the SIO MSSH model (4 cm).






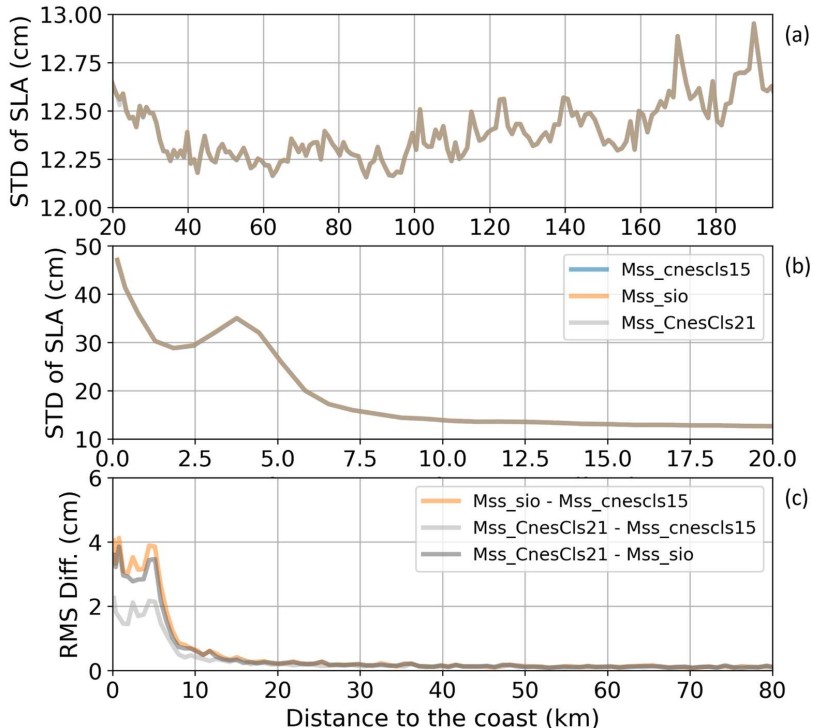

**Figure 9:** (a) and (b): Global mean of standard deviation values (in cm) of the SLA along all Jason-2 tracks for the period 27/09/2013 to 02/12/2016 when applying different MSSH solutions . Results are represented as a function of the distance to the coast (in km) between 200 km and 20 km (a) and between 20 km and 0 km from the coast (b). (c): Global standard deviation of the differences of standard deviation

values (in cm) of the SLA when applying different MSSH solutions between 80 km and 0 km from the coast.

### 4.5 Altimeter range and SSB

The accuracy of the altimeter range is directly related to the retracking method used. The latter consists in an analytical model fitted to the satellite waveform in order to derive geophysical information (the so-called retracking), including the range, the

significant wave height and the wind speed. The SSB aims to correct the error in the satellite altimetry sea level measurements that is due to the presence of ocean waves at the ocean surface (Tran et al., 2022). Its estimation is based on empirical models (Gaspar et al., 2002) computed from the significant wave height and wind speed estimated during the retracking step. The altimeter range and the SSB used in the SLA calculation are then necessarily dependent since derived from the same analytical model. In the coastal zone, they are both impacted by the presence of more complex altimetry waveforms at about 10-15 km

from the shore, due to land contamination. This results in noisier fields at the output of the retracker (Andersen and Scharroo,



2011; Gommenginger et al., 2011). The SSB computation is also more complex in coastal areas because of the changing shape of the wave and wind fields (Dodet et al., 2019). In this section, these two SLA components (i.e. range and SSB) will first be analyzed together. We will then focus on the SSB alone for which several calculation methods exist for a given retracker.

The MLE4 2D retracking algorithm (Thibaut et al., 2010) is the standard method used for the operational processing of LRM altimetry waveforms over the ocean. However, when approaching the coast, as mentioned above, the presence of signals coming from land in the altimetry waveforms impacts the ability of MLE4 to retrieve accurate geophysical variables. Different algorithms have been developed during the past years to improve the SLA data retrieval in nearshore areas. This is the case of the ALES (Passaro et al., 2014) and Adaptive (Poisson et al., 2018) retracking algorithms. In this section, we will analyze the differences in SLA obtained when using the range and SSB derived from these three retrackers (MLE4, ALES and Adaptive) and the way they behave when approaching the coastline. Here we only show results for Jason-3 because, unfortunately, it turned out that a significant number of Jason-2 cycles were missing in the Adaptive dataset.

First, we compare the SLA estimates computed with the range-SSB couple associated with each of the retracking algorithms considered in the round robin. For the MLE4 retracking, the SSB version considered here is the 2D dataset at 1Hz (GDR standard) interpolated at 20 Hz. For the Adaptive and ALES retracking algorithms, it is the 2D SSB solution directly computed at 20 Hz. For Jason-3, the global mean of the STD of SLA is 14 cm, 13.8 cm and 14.1 for MLE4, Adaptive and ALES, respectively (for more details and plots, see https://www.aviso.altimetry.fr/en/data/products/sea-surface-height-products/global/altimetry-innovative-coastal-approach-product-alticap/roundrobin-reports.html). However, when we represent the STD of SLA for the three retrackers as a function of the distance to the coast (Figure 10 a and b), we see that these average numbers can mask significant spatial differences, particularly in the last 10 km to the coast. Note that the statistics associated with MLE4 are not completely comparable to those of the other retracking algorithms below 10 km because the number of data available at the ouput of the MLE4 retracker drops by about 20%, whereas the number of data available at the output of ALES and Adaptive remains stable up to about ~4-5 km from the coast.

Here again, the spread between the SLA solutions obtained with these three retracking algorithms (Figure 10.c) clearly increases when approaching the coast, reflecting an increase in the SLA uncertainty associated to the range-SSB couple. The associated STD values of the differences are below 0.5 cm in the open ocean up to 60 km from the coast, then they range between 0.5 and 1.5 cm between 10 and 60 km, and they finally increase up to 4 cm in the last 10 km.



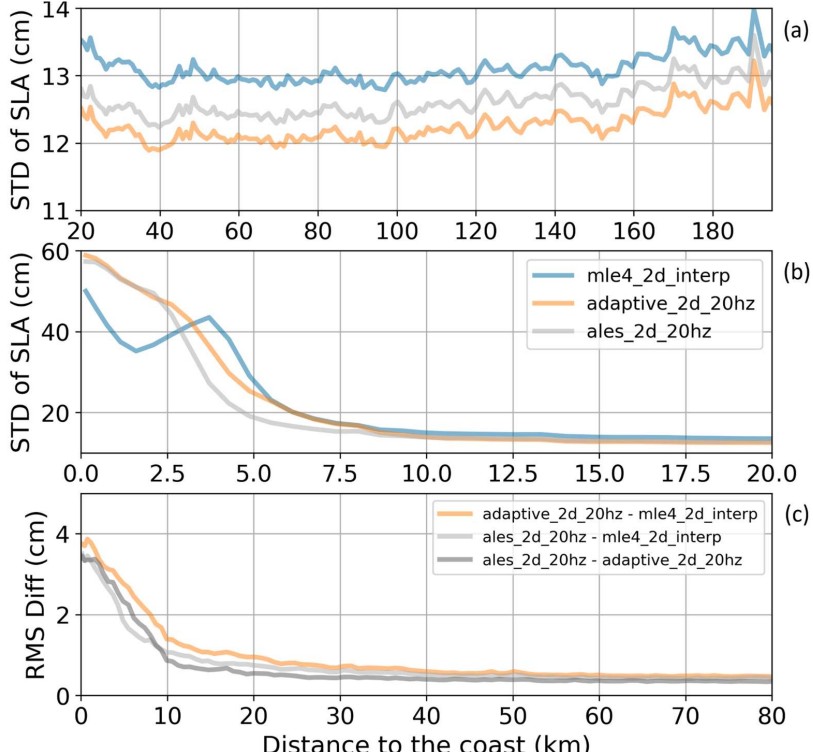

**Figure 10:** (a) and (b): Global mean of standard deviation values (in cm) of the SLA along all Jason-3 tracks for the period 17/02/2016 to
22/02/2019 when applying different retracking solutions for the range and the SSB corrections. Results are represented as a function of the
distance to the coast (in km) between 200 km and 20 km (a) and between 20 km and 0 km from the coast (b). (c): Global standard deviation
of the differences of standard deviation values (in cm) of the SLA when applying different retracking solutions for the range and the SSB
between 80 km and 0 km from the coast.

We now focus on the SSB correction. In the operational SLA processing, the reference correction is the MLE4 SSB

calculated at 1Hz and then interpolated at higher rate (20Hz). Passaro et al (2018) showed that the computation of the SSB

correction directly at 20Hz improves the accuracy of the SLA data. Moreover, according to Tran et al. (2021), by using a 3D

version of the SSB correction instead of the standard 2D version, we obtain an SLA variance reduction for the high-frequency

signals. Here, we will inter-compare the impact of 5 SSB solutions on the SLA computation as we approach the coast (Figure

11). Three are associated with MLE4: the 2D version of the SSB computed at 1Hz and interpolated at 20Hz, the 2D version of

the SSB directly computed at 20Hz, and the 3D version of the SSB computed at 20Hz (Figure 11 a,b,c). The two other solutions

are associated with the Adaptive retracker: the 2D version of the SSB computed at 20Hz and the 3D version of the SSB



computed at 20Hz (Figure 11 d,e,f). Note that we do not consider the ALES solution here as we want to focus on the impact of the SSB, separately from that of the range. Hence we can only inter-compare SSB solutions associated with a given retracker
(only one ALES SSB solution available).

For the entire study area considered, for MLE4, the mean STD of the SLA obtained is 14 cm for the SSB 2D 1Hz, 13.2 cm for the SSB 2D 20Hz and 12.9 cm for the SSB 3D 20Hz (see https://www.aviso.altimetry.fr/en/data/products/sea-surface-height-products/global/altimetry-innovative-coastal-approach-product-alticap/roundrobin-reports.html). For Adaptive, it reaches 13.8 cm and 12.8 cm for the SSB 2D 20Hz and SSB 3D 20Hz, respectively. These results show the strong impact of
the SSB processing on the SLA estimates on a global scale, and particularly the consequence of the interpolation from 1 Hz to 20 Hz, as it is commonly done when using the operational altimetry products, on the resulting STD values.

If we represent the STD of SLA for the various SSB solutions as a function of the distance to the coast (Figure 11 a, b, d and e), we can see differences of the order of 1 cm between the solutions in the open ocean up to about 10 km from the coast. In the last 10 km, larger discrepancies are observed among the curves, showing different shapes as the values of STD of SLA
increase close to the coast.

The spread between the SLA solutions obtained with the various SSB corrections (Figure 11 c and f) provides an estimate of the uncertainties associated with the SSB, ranging from 1.8 to 6.2 cm in the last 10 km to the coast. This means that the available SSB solutions strongly disagree very close to the coast. However, the largest coastal discrepancies (more than 6 cm) are observed between the oldest (MLE4 SSB 2D 1Hz interpolated at 20 Hz) and the newest (MLE4 SSB 3D directly
computed at 20 Hz) approaches. When considering the 2D and 3D SSB solutions directly computed at 20 Hz, the maximum discrepancies are lower, in the order of 3.5 to 4 cm for both the MLE4 and Adaptive algorithms. The most recent approaches to calculating SSB thus appear to reduce the SLA uncertainty, and we can expect some further reductions in the future as works are still on going to improve this correction.



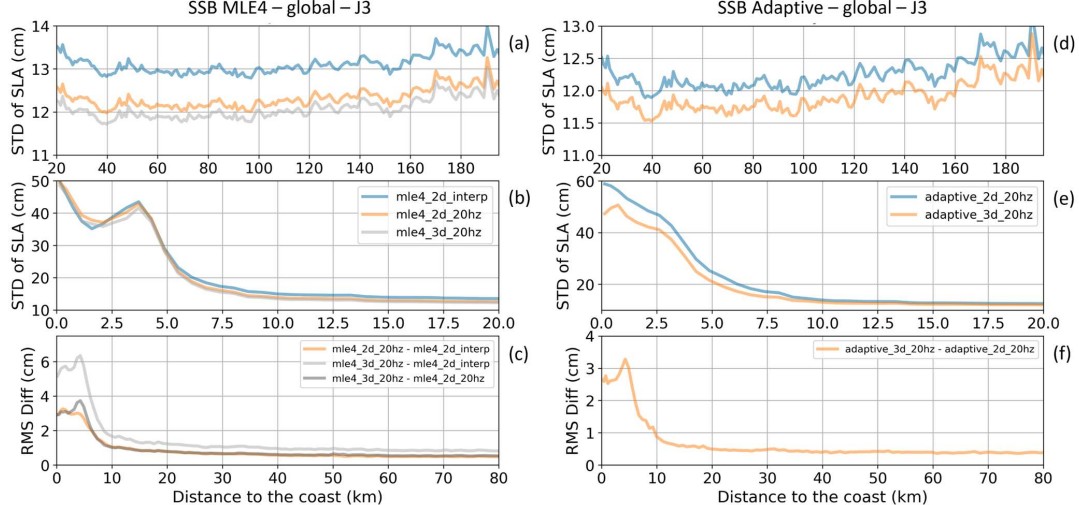


**Figure 11:** (a), (b), (d), (e): Global mean of standard deviation values (in cm) of the SLA along all Jason-3 tracks for the period 17/02/2016 to 22/02/2019 when applying different solutions for the SSB corrections associated with the MLE4 (a and b) and Adaptive (d and e) retracking algorithms. Results are represented as a function of the distance to the coast (in km) between 200 km and 20 km (a and d) and between 20 km and 0 km from the coast (b and e). (c and f): Global standard deviation of the differences of standard deviation values (in cm) of the SLA when applying different SSB corrections associated with the MLE4 (c) and Adaptive (f) retracking algorithms between 80 km and 0 km from the coast.

## 4.6    Synthesis of the results

Assuming that the spread of SLA values obtained by changing the calculation algorithms provides an estimate of the associated
SLA uncertainty, we summarize in Table 2 the main results found in this study for the different SLA components. Beyond the near-coastal region, the biggest contributors to SLA uncertainties are the SSB and the range, both associated with the retracker algorithms, generating an uncertainty of about 1 cm. Then comes the tidal correction with an associated uncertainty between 0.5 and 1 cm, depending on the tidal models that are considered (in particular, the extent of the tidal model spectra is a key player in the estimation of the associated uncertainties, as more tidal signal is removed from the altimeter SLA when using
models with a richer spectrum). The MSSH also contributes in the order of 0.5 cm to the SLA uncertainties in the open ocean. For the other components (ionospheric and wet tropospheric corrections), the solutions tested generated difference envelopes of less than 0.3 cm.

For all components, the SLA uncertainties start to slightly increase at some distance to the coast (75 km for the tides, 60 km for the range and SSB, 40 km for the ionospheric and wet tropospheric corrections, 20 km for the MSSH), reaching
between 0.5 cm (ionospheric correction) and about 2 cm (SSB, tides) at about 10 km from the coast. The largest SLA



uncertainties associated with these components are all observed in the last 7.5 to 10 km to the coast, where the spread between all the SLA estimates strongly increases, reaching several centimetres for all the components, from about 2 cm for the ionospheric correction to about 6 cm for the SSB.

In addition, these average uncertainty values mentioned in Table 2 hide significant spatial variations, with levels that can be
locally higher, for instance in areas of strong bathymetric gradient for the MSSH and of large tidal amplitudes or complex features for the tidal correction (Figure 7). For the wet troposphere correction, this result is true provided that a recent version of the radiometric correction is used.

    Finally, when we get very close to the coast, at around 10-15 km from land, the availability of the SLA components can play an important role in the statistics, with for instance some artificial drops in the STD of SLA due to a lower amount of
available data, like can be noticed for the altimeter range and SSB (Figure 10.b), with possible differences of several tens of cm that are beyond the amplitude of the oceanographic signals we want to observe. The choice of the retracker algorithm thus becomes really critical if we want to use altimeter data in the nearshore area.

| SLA component | Uncertainty estimate | Coastal zone impacted |
|---|---|---|
| **Ionospheric correction** | 0.7 – 2.8 cm<br>0.2 – 0.7 cm<br>< 0.2 cm | 0 – 10 km<br>10 – 40 km<br>> 40 km |
| **Wet tropospheric correction** | 0.5 – 1.7 cm<br>0.3 – 0.5 cm<br>< 0.3 cm | 0 – 7.5 km<br>7.5 – 40 km<br>> 40 km |
| **Ocean tide correction (*)** | 1 (2) – 4 (5) cm<br>0.5 (1) – 1 (2) cm<br>0.5 (1) cm | 0 – 10 km<br>10 – 75 km<br>> 75 km |
| **MSSH** | 1 – 4 cm<br>0.5 – 1 cm<br>< 0.5 cm | 0 – 8 km<br>8 – 20 km<br>> 20 km |
| **Retracking (range + SSB)** | 1.5 – 4 cm<br>0.5 – 1.5 cm<br>< 0.5 cm | 0 – 10 km<br>10 – 60 km<br>> 60 km |
| **SSB correction** | 1.8 – 6.2 cm<br>1 – 1.8 cm<br>< 1 cm | 0 – 10 km<br>10 – 60 km<br>> 60 km |

**Table 2:** SLA components included in the study (column 1), maximum spread of the differences in the STD(SLA) (uncertainty estimate) observed when we change the solution for this component (column 2) and oceanic region where these differences are observed (column 3).
(*) For the ocean tide correction, the values in brackets correspond to uncertainty estimates considering the EOT20 model, while the other values correspond to the FES2014 and regional models only.



It is also important to note that these uncertainty estimates are interlinked from one component to the other and not independent from each other, as most of them are also based on satellite altimetry observations (e.g. SSB, MSSH, tidal models...). The total uncertainty associated with all the components thus cannot be estimated as the direct sum of the uncertainty for each component.

Of course, we cannot be sure that these results reflect the estimate of how far the SLA obtained may be from the true SLA value because no measure of truth exists. Here, the SLA uncertainty is estimated through the analysis of the STD of SLA obtained for each solution. In altimetry, this is a classical diagnosis of the algorithm performance, considering that a solution performs well when it reduces the variability in the SLA. Nevertheless, it is possible to have small uncertainty estimates because all the considered solutions to compute the SLA are close to each other, and still observe large errors in the SLA. As this study covers a wide range of algorithms, including the most recent and efficient algorithms available today to compute altimetry SLAs, it probably represents the best we can do today in estimating altimeter uncertainties.

## 5    Conclusion

The contribution of satellite altimetry to scientific advances in the field of ocean dynamics is unique in the history of Earth observation from space (International Altimetry Team 2021). It is now critical to improve and understand sea level observations from altimetry in coastal areas, so that they can play a major role in coastal oceanography. This requires an understanding of the current sources of uncertainty in the data, and then their reduction. In this study, we take advantage of the availability of several algorithms for most of the terms/corrections used in the calculation of the altimeter SLA to estimate the uncertainties associated when approaching the coast. A round robin exercise testing a total of 21 solutions for retracking radar altimeter data, correcting sea surface heights and finally deriving sea level variations has been performed. All solutions are evaluated through the same metrics, at both global and regional scales, and as a function of the distance to the coast. The results show that SLA uncertainties remain low and stable beyond 40-60 km from the coast, making them very reliable to use in this area. Within this distance, uncertainty values start to increase gradually. They can still be used with caution, especially if the ocean signal studied is larger than a few centimeters, up to 10 km from the coast. Then, they reach levels of magnitude above most ocean dynamics signals. In terms of origin, uncertainties in ocean tide models and in mean sea surface height models significantly contribute to the coastal SLA uncertainty budget in some regions. The altimeter range and the SSB appear to be large contributors to SLA uncertainties in the open ocean but within 10 km off the coastline, they become the limiting factor in the use of altimetry data. If the result is that coastal users should give preference to altimetry data sets based on retrackers developed for coastal objectives, such as Adaptive and ALES, the remaining uncertainty levels underline the importance of further improvements in this domain.

Finally, it is important to keep in mind that the findings of this study are intrinsically related to the algorithms currently available to compute the altimeter SLA. They may significantly evolve in the future thanks to new methods and algorithms. We believe this work is important to better understand and characterize the current sources of errors and uncertainties in the



altimetry measurements in coastal sea areas. This is the reason why the results obtained have already been transferred to the CNES operational computing centre. In parallel, based on this study, we have started to work on the computation of sea level uncertainties that can be added to coastal altimetry products. This should greatly facilitate the use of these datasets by a wider scientific community. Note that even if this work was carried out with LRM altimetry data, part of the conclusions should also contribute to modern altimetry techniques such as SAR and SARin, as all satellite altimetry missions share some common

correction terms, such as tidal and MSSH models for example. Even with their increased observational capabilities, which are favorable for monitoring coastal zones, the way these new types of altimetry observations are processed and the methodologies used to calculate the various geophysical corrections remain critical steps to derive accurate and precise geophysical information.

*Author contributions.* FB, MC, FBC and MIP initiated and designed the study. WF performed all the analyses under supervision from all authors. FB led the paper and wrote and structured the manuscript. All authors discussed the analyses and provided comments and corrections to the text.

*Competing interests.* The authors declare no competing interests.


*Acknowledgements:* This study was funded by the Centre National d'Etudes Spatiales (CNES) which is gratefully acknowledged for its support. It was only made possible because the algorithms tested in this study being made available by the teams developing them. We would like to thank them for this and in particular M. Passaro from Technical University Munich and J. Fernandes from University of Porto. We wish also to acknowledge the contribution of the services that make

the tide gauge data available: CMEMS, Refmar, ISPRA, BODC, UHSLC and BOM.

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
