# Peer review of "Understanding uncertainties in the satellite altimeter measurement of coastal sea level: insights from a round robin analysis."

_EGUsphere, 2024_

## Community Comment (CC1)

**Round Robin Assessment of altimetry algorithms for coastal Sea Surface Height data**

Florence Birol; François Bignalet-Cazalet; Mathilde Cancet; Jean-Alexis Daguze; Yannice Faugère; Wassim Fkaier; Ergane Fouchet; Fabien Léger; Claire Maraldi; Fernando Niño; Marie-Isabelle Pujol; Ngan Tran; Pierre Thibaut

Compare different algorithms used in the SSH computation and gain insights into their ability to contribute to obtaining quality data in the « coastal » band.

**Why?**

1.  **Investigate which component (range, correction, MSSH) is the most limiting near the coast**
2.  **Define a baseline for the generation of a new global coastal SLA product**

OSTST 2022

**Specifications**

- **Altimetry:** LRM (focus on long time series)

- **Variable** : SLA

- **Frequency**: 20 Hz

- **Missions**: Jason-2 & Jason-3

- **Period** : 3 years for each mission (111 cycles)

- **Zone**: Global coastal ocean (0-200km) + regional.
3 regions: Mediterranean Sea, NEA, Eastern Australia

[Figure]

[Figure]

[Figure]

**Parameters considered**

Selected because available at global scale for both Jason-2 & Jason-3 and over the period analysed

| SLA component | List of algorithms | |
|---|---|---|
| **Range** | MLE4 (REF), Adaptive, ALES | → 3 solutions |
| **Ionospheric correction** | Dual frequency filtered (REF), GIM | → 2 solutions |
| **Wet tropo correction** | Radiometer (REF), ECMWF, GPD+ | → 3 solutions |
| **Ocean tide** | DTU16, EOT20, FES2014 (REF: regular grid, unstructured mesh), GOT4.10, TPX09, CNES Regional models (NEA, Med, Australia, Arctic) | → 6 solutions |
| **SSB** | MLE4 2D 1Hz (REF), MLE4 20Hz, MLE4 3D 20Hz, Adaptive 2D 20Hz, Adaptive 3D 20Hz, solution ALES 20Hz | → 6 solutions |
| **MSSH** | CNES15 (REF), SIO, CNES22 | → 3 solutions |

Reference: standards used today in the GDRs to compute the SSHA parameter, as well as in the L3/L4 SLA products

**Total:** 22 algorithms tested

**A framework for assessing performance**

- **Intercomparison between the different algorithms for each SLA component**

Objective: for each algorithm, measure the internal consistency compared to the reference solution and its performance in terms of SLA data availability and SLA variance reduction, as a function of distance to the coast

Histograms, maps of MEAN and STD, % of data as a function of distance to the coast, MEAN and STD as a function of distance to the coast. GLOBAL + REGIONAL

- **External data comparison using in-situ measurements:**

Objective: use independent tide gauge data to measure the impact of each algorithm on the SLA calculation.

Statistics (correlation, RMSD), SLA data availability at local scale, Taylor diagrams. REGIONAL

- **Intercomparison between 2 altimetry missions:**

Objective: for each algorithm, measure the consistency of all the results between different altimetry missions

All the reports mentioned above

A specification document will be made freely available

**Results – wet tropo**

**Analysis at global scale - Jason-2**

**STD(wet) as a function of distance to the coast for the 3 solutions**

[Figure]

**Same results for Jason-3**

[Figure]

**STD(GPD) – STD(Rad) in cm**

→ STD: Differences between the 3 solutions < 0.3 cm
→ Differences between RAD & GPD solutions very small up to 7-8 km to the coast

**Results – wet tropo**

**Analysis at global scale - Jason-2**

**STD(SLA) = f(dist_to_coast)**

[Figure]

**STD(SLA_wet1) - STD(SLA_wet2)**

[Figure]

→ Impact on STD(SLA) < 0.1 cm near the coast at global scale, but can be slightly larger locally

R1: Concerning RAD, the results are highly related to the processing version
R2: Impact on the long term SLA evolution not included in this RR exercise

**Results – ocean tide**

**Analysis at regional scale - Jason-3**

Example of the NEA region

6 model families compared with tide gauge and altimetry, assessing **DAC compatibility :**

**DTU, EOT20, FES2014(g/u), GOT4.10, TPX09, RegAT(g/u)**

→ Best performance for FES2014 unstructured mesh + regional models

**See also poster COA2022_004**

*Residual Sum of Squares = sea level variance not explained by the ocean tidal model (REF: altimetry or TG)*

**Results – Range + SSB**

**Analysis at global scale - Jason-3**

**Nb of SLA = f(dist_to_coast)**

[Figure]

**SLA data availability**

Total number of cycles : 111

→ Compared to other retrackers, MLE4 stalls at 10km to the coast

→ Adaptive and ALES both recover significantly more data within 10 km of the coast

→ In terms of number of coastal SLA data, ALES is the most efficient algorithm

**Results – Range + SSB**

**Analysis at global scale - Jason-3**

**STD(SLA) = f(dist_to_coast)**

[Figure]

**SLA Variance reduction**

**Important remarks:**
- the HFA correction (Adaptive) and its equivalent for ALES are not used in this study
- For each solution, the SSB used changes depending on the retracker (2D solutions used for MLE4 & Adaptive)

→ Differences observed: ~1.5 cm offshore, ~15 cm at 4 km

→ **15 km < dist < 200 km** : the adaptive retracker gives the lowest values in terms of STD(SLA)

→ **2 km < dist < 15 km** (if we forget MLE4 not significant because of data loss): the ALES retracker gives the lowest values in terms of STD(SLA)

→ ALES generally slightly better in terms of statistics at the tide gauges (not shown)

**Round Robin Results: summary**

**Objective 1: Investigate which component is the most limiting near the coast**

**Differences observed near the coast in terms of STD(SLA), according to the SLA component: first analysis**

| SLA component | Difference Amplitude | Coastal zone with differences | Comment |
|---|---|---|---|
| **Range** | 1-10 cm

~1 cm | 10-15 km

0-200 km | Very important in the first 10 km, especially for MLE4

Impact also (but less) further offshore |
| **Ionospheric correction** | 0.2 cm | Not specific to the coastal zone | Dual frequency solution: loss of points due to filtering, especially on J3 |
| **Wet tropo correction** | < 0.5 cm | 7-8 km | For the radiometer, the result depends on processing versions |
| **Ocean tide** | < 5 cm | 10 km | < 2 cm beyond 5 km
But results very heterogeneous spatially |
| **SSB** | 1-15 cm
~1 cm | 10-15 km
0-200 km | To be refined by removing impact of the retracker |
| **MSSH** | 0-5 cm | ~30-50 km | Impact < 0.1 cm offshore and < 1 cm up to 7-8 km |

**Round Robin Results: summary**

**Objective 1: Investigate which component is the most limiting near the coast**

**Differences observed near the coast in terms of STD(SLA), according to the SLA component: first analysis**

| SLA component | Difference Amplitude | Coastal zone with differences | Comment |
|---|---|---|---|
| **Range** | 1-10 cm | 10-15 km | Very important in the first 10 km, especially for MLE4 |
| | ~1 cm | 0-200 km | Impact also (but less) further offshore |
| **Ionospheric correction** | 0.2 cm | Not specific to the coastal zone | Dual frequency solution: loss of points due to filtering, especially on J3 |
| **Wet tropo correction** | < 0.5 cm | 7-8 km | For the radiometer, the result depends on processing versions |
| **Ocean tide** | < 5 cm | 10 km | < 2 cm beyond 5 km
But results very heterogeneous spatially |
| **SSB** | 1-15 cm
~1 cm | 10-15 km
0-200 km | To be refined by removing impact of the retracker |
| **MSSH** | 0-5 cm | ~30-50 km | Impact < 0.1 cm offshore and < 1 cm up to 7-8 km |

**Round Robin Results: summary**

**Objective 2: Define a baseline for the generation of a new global coastal SLA product**

**Baseline selected considering algorithms available on J2&3 and results on the whole [0-200 km] coastal band**

| SLA component | List of algorithms | |
|---|---|---|
| **Range** | MLE4, **Adaptive**, ALES | → NEW |
| **Ionospheric correction** | Dual frequency filtered, **GIM** | → NEW |
| **Wet tropo correction** | Radiometer, ECMWF, **GPD+** | → NEW |
| **Ocean tide** | GOT4.10, FES2014 regular grid, **FES2014 unstructured mesh, CNES regional models (NEA, Med, Australia, Arctic)**, TPXO9v4, EOT20 | → NEW |
| **SSB** | MLE4 2D 1Hz, MLE4 20Hz, MLE4 3D 20Hz, **Adaptive 2D 20Hz**, Adaptive 3D 20Hz, solution ALES 20Hz | → NEW |
| **MSSH** | CNES15, SIO, CNES22 | → Still under analysis |

**Many changes!**

**Conclusion**

- ❑ Still a bit of work to refine the analysis
- ❑ Numerous reporting tools available; a summary of the protocol and main results will be published soon.
- ❑ Many CNES/LEGOS/CLS/Noveltis exchanges of expertise in the technical and scientific domain… and now an established working group
- ❑ A new global product (L2P) covering the [0-500 km] coastal band and the Jason-3 mission planned for April 2023 (V1)
- ❑ Recommendations to space agencies in terms of studies to be funded (range, corrections)
- ❑ Next news at the Coastalt Workshop

**All comments / questions / requests are welcome!!!**

---

## Author Response (AR1)

**General answer to Reviewers:**

We first would like to thank the two reviewers for providing very constructive suggestions in order to improve our manuscript. We also appreciate Marcello Passaro's comment, thank you. A point-by-point answer follows.

\*\*\*\*\*\*\*\*\*\*\*\*\*\*\*\*\*\*\*\*\*\*\*\*\*\*\*\*\*\*\*\*\*\*\*\*\*\*\*\*\*\*\*\*\*\*\*\*\*\*\*\*\*\*\*\*\*\*\*\*\*\*\*\*\*\*\*\*\*\*\*\*\*\*\*\*\*\*\*\*\*\*\*\*\*\*\*

**Answer to Reviewer 1**

General points:
The aim of the paper is to analyse the source of "uncertainties" in sea level measurements from radar altimetry data, based on the dispersion of sea level anomaly values obtained by the different algorithms and corrections.
The paper is well written, with all aspects of the analysis clearly explained.
The paper should make it clear that this analysis is of uncertainties in the measurement of sea level anomaly resulting from different processing algorithms and with different sources of corrections, as distinct from uncertainties due to random errors in the measurement of sea level, e.g. due to small scale variability.

**Answer**: Thanks a lot for this comment. We have changed the abstract slightly as well as the introduction to clarify this point:
In the abstract, now: "Here, we take advantage of the recent availability of many new algorithms developed for altimetry sea level computation **to quantify and analyze the uncertainties associated with the choice of algorithms** when approaching the coast."
In the introduction now: " The main objective of this paper is to take advantage of them to better understand the sources of uncertainties **linked to the processing algorithms** in the sea level computation when approaching the coast."

The discussion should make clear this analysis applies specifically to LRM data, and not to SAR Altimetry data.

**Answer**: We have added a dedicated sentence in the conclusion: "**We are focusing on LRM altimetry, which has the longest data history and the largest number of processing algorithms available**."

I would like to have seen some discussion on the reasons behind the differences in performance between the different algorithms and corrections. Are there potential physical reasons?

**Answer**: We have added some sentences in the conclusions (as well as a reference to a recent study). Now (text added in bold):
"In terms of origin, uncertainties in ocean tide models and in mean sea surface height models significantly contribute to the coastal SLA uncertainty budget in some regions. **About tidal models, despite major progress, the spatial resolution remains inadequate to take account of the dynamics of the most coastal tide (Hart Davis et al., 2024). Concerning MSSH solutions, they are still poorly constrained near the coast due to the lack of SLA data used to calculate them and their poorer quality (Pujol et al., 2018).** The altimeter range and the SSB appear to be large contributors to SLA uncertainties in the open ocean but within 10 km off the coastline, they become the limiting factor in the use of altimetry data. **This is due to the complexity of radar echoes near the coast, which makes them much more difficult to**

**model.** If the result is that coastal users should give preference to altimetry data sets based on retrackers developed for coastal objectives, such as Adaptive and ALES, the remaining uncertainty levels underline the importance of further improvements in this domain."

OS questions:

Does the paper address relevant scientific questions within the scope of OS?

Yes – the main question – to gain a better understanding of uncertainties in satellite altimeter measurement of sea level anomaly is relevant to, and lies within the scope of, OS

Does the paper present novel concepts, ideas, tools, or data?

The paper introduces and compares results from new processing algorithms. Thus the data and analysis are new.

Are substantial conclusions reached?

The conclusions provide new insight into the uncertainties in different aspects of the calculation of sea level anomaly from satellite data. There are no new insights into the physics of variability in sea  level anomaly

Are the scientific methods and assumptions valid and clearly outlined?
Are the results sufficient to support the interpretations and conclusions?
Is the description of experiments and calculations sufficiently complete and precise to allow their reproduction by fellow scientists (traceability of results)?

Yes -  the methods are valid, clearly explained and support the conclusions. The approach can be reproduced, and the data accessed, on the information provided

Do the authors give proper credit to related work and clearly indicate their own new/original contribution?

Yes – the references are relevant and appropriate

Does the title clearly reflect the contents of the paper?

I would prefer that the title made it clear that it was the uncertainties in the measurement of sea level anomaly that was being assessed, e.g.:

Understanding uncertainties in the satellite altimeter measurement of coastal sea level altimetry data: insights from a round robin analysis.

**Answer:** Thanks a lot for this suggestion. We agree and have changed the title accordingly.

Does the abstract provide a concise and complete summary?
Is the overall presentation well structured and clear?
Is the language fluent and precise?

The abstract and main text are well written and clear. Some specific recommendations for clarification of language have been provided.

**Answer**: Thank you for this meticulous proofreading and the corrections made. We have included them into the manuscript (see below).

Are mathematical formulae, symbols, abbreviations, and units correctly defined and used?

Yes

Should any parts of the paper (text, formulae, figures, tables) be clarified, reduced, combined, or eliminated?

Some specific recommendations for clarifications have been provided

Are the number and quality of references appropriate?

In general yes – some references are missing / incorrect.

**Answer**: Here again, your corrections have been included in the revised version of the manuscript (see below).

Is the amount and quality of supplementary material appropriate?

Not applicable

Specific comments

Section 2.1 General Goals.

P3 Line 70 Re the focus on LRM data. Should note then that this analysis of uncertainties is specific to LRM data. SAR Altimeter Delay Doppler processed data will have different characteristics.

**Answer**: We have added the following sentence at the end of the 1rst paragraph of section 2.1: "Note then that the results presented below are specific to LRM altimetry data."

P3 Line 93 Laignel et al, 2022 – not in references.

**Answer**: It has been corrected for Laignel et al., 2023 and added in the reference.

P4 Line 105 "uncertainties in sea level data". Need to be careful not to generalise – this analysis only provides information on variability in sea level data from different processing approaches, not directly on variability in the original sea level data.

**Answer**: Right. Now: "Finally, note that the focus of this round robin study is the comparison of different processing solutions in order to gain insight about the **associated** sources of uncertainties in sea level data when approaching the shoreline."

Section 2.2 Overview of the selected algorithms

P5 line 127. Some missing words: "were also discarded because they are considered as very accurate ….."

**Answer**: Done.

P5 line 134 "aforementioned article" – replace with Cazenave et al., 2022 to avoid any uncertainty.

**Answer**: Done.

Section 2.3 Tide Gauge Data

Line 166, 168 – I get an error for the refmar URL – just give the main French URL

**Answer**: Corrected.

Section 3 Methodology

P9 Line 212: "as the density of altimetry points is higher close to the coasts due to the presence of islands and to the tracks configuration."
I do not understand this statement. The density of points cannot be higher due to the presence of islands. If anything it should be lower. Please rewrite this sentence.

**Answer**: When considering all altimetry points of our global coastal dataset, there are statistically more points at a short distance to the coast than at a long distance, because of the presence of islands and the configuration of the tracks (for example, a track can be parallel to the coast over kilometers at a distance of 5 km to the coast). If we consider regular bins in terms of distance to the coast (e.g. every kilometer) we obtain a repartition that is very heterogeneous, with much more points in the bins of the last 5 km to the coast than in the bins offshore. Hence we chose to create bins based on a fixed number of points instead to ensure consistent statistics.

Now (text added/changed in bold):
To ensure robust global or regional statistics, we considered a fixed number of altimetry points in each bin, with the bin size varying from about 300 m at the coast to 1.2 km at 200 km from the coast, **as the distribution of altimetry points as a function of the distance to the coast shows higher density of points close to the coasts** due to the presence of islands and to the tracks configuration.

Section 4.1  Results / Ionospheric Correction

P11 Figure 3. Caption refers to blue and green lines. Figure has blue and orange lines.

**Answer**: Corrected.

P11 line 259. Add "in SLA" to the end of this sentence – "as are potentially the uncertainties in SLA"

**Answer**: Corrected.

Section 4.2  Wet Tropospheric Correction

P12 line 268 – "Due to the large space-time variability of this correction (0-50 cm), the latter is considered the best option.  Should provide a reference for this statement – also, is this specifically over ocean?

**Answer**: A reference has been added (**Obligis et al., 2011**). The problem is not specific to the ocean, but the radiometer only gives a correct correction on this type of surface. For applications on continental waters, we need to use the model solutions.
Now: "Due to the large space-time variability of this correction (0-50 cm), the latter is **generally considered the best option over the ocean (Obligis et al., 2011)**."

P13 line 283 Use "pair" instead of "couple" in "…differences of STD of SLAs for each couple of SLA solutions…"

**Answer**: Done.

Section 4.3 Ocean Tide Correction

P16 343-344. I find this sentence confusing: "In reddish (blueish) regions, FES2014b (the regional solution) decreases the STD of SLA more significantly."   Does the regional model (FES2014b) decrease the STD of the SLA more significantly in both red and blue coloured regions?

Please rewrite to make the point clearly.

**Answer**: Done. Now: "**In reddish regions, FES2014b decreases the STD of SLA more significantly; in blueish regions, it is the regional solution that reduces the SRD of SLA the most**."

P16 line 353. Suggest to replace "if" by "although", and add "such"  i.e. : "These results illustrate that althoughvery significant progress has been made since studies such as Ray (2008), large uncertainties remain…"

**Answer**: Done.

P17 line 363. "However, if this result can probably be extrapolated to the whole Mediterranean Sea, which is characterized by small tidal amplitudes except in a few areas (Adriatic Sea, Gulf of Gabes), it should be qualified for the Australia region, as it strongly depends on the tide gauge stations." It is not clear to me what this sentence means. What is "the result" that can be extrapolated? Is it that the FES2014b regional is the TG model that gives the highest correlation and lowest RMS? If so, this should be explicitly stated.

**Answer**: Right. This sentence has been removed. Now: "**The Mediterranean Sea is a micro-tidal zone. However, concerning the Australia region, this result could be affected by the choice of tide gauge stations used for the analysis**. Indeed, …"

Section 4.4 Mean Sea Surface Height

P18 line 376 Andersen and Knudsen, 2009 is not included in the references.

**Answer**: Reference added.

**Answer**: The correct reference is Pujol et al., 2018. It has been corrected.

**Answer**: The correct reference is Schaeffer et al., 2023. It has been corrected.

P20 line 419 "For the Adaptive and ALES retracking algorithms, it is the 2D SSB solution directly computed at 20 Hz." Replace "it is" by "we consider"

**Answer**: Done.

P20 line 426 Could an additional panel be added to Figure 10 to illustrate the point that the MLE4 retracker provides fewer valid data solutions than the other retrackers?

**Answer**: Done, we have added the additional panel on Figure 10.

P20 line 428-9 "Here again, the spread between the SLA solutions obtained with these three retracking algorithms (Figure 10.c) clearly increases when approaching the coast, reflecting an increase in the SLA uncertainty associated to the range-SSB couple." It is not clear what "couple" means here?
Maybe rephrase, e.g.
"Here again, the spread between the SLA solutions obtained with these three retracking algorithms (Figure 10.c) clearly increases when approaching the coast, reflecting an increase in the SLA uncertainty associated to uncertainties in range and SSB."

**Answer**: Done.

P23 Line 479-480 "Assuming that the spread of SLA values obtained by changing the calculation algorithms provides an estimate of the associated SLA uncertainty,…"
This text could be interpreted to read that the spread of SLA values from the different calculations is representative of geophysical variability in SLA. To avoid this interpretation I suggest to remove the first part of the sentence so that it starts "We summarize in Table 2 the main results…..

**Answer**: Done.

P23 line 481. And similarly I'd suggest to change

"Beyond the near-coastal region, the biggest contributors to SLA uncertainties are the SSB and the range, both associated with the retracker algorithms, generating an uncertainty of about 1 cm."
to
"Beyond the near-coastal region, the biggest contributors to uncertainty in the LRM altimeter estimate of SLA  are the SSB and the range, both associated with the retracker algorithms, generating an uncertainty of about 1 cm."

**Answer**: Done.

Rest of this section.

In general I would recommend to replace "SLA uncertainties" with "uncertainties in the estimated SLA"

**Answer**: Done.

P25 line 512

I think some more precise language is required.
"Of course, we cannot be sure that these results reflect the estimate of how far the SLA obtained may be from the true SLA value because no measure of truth exists."
The basis of the discussion is that the analysis provides an estimate of the range of possible values of SLA using different approaches to the calculation. Thus, the argument of this paper goes, this is a proxy for an estimate of precision. It is not aiming to assess accuracy (how close the measurement is to the actual value – which is only available at tide gauges).
The discussion should make the distinction between precision and accuracy, and also note the range of different estimates is due to different approaches to the calculation and not necessarily representative of the individual precision of any single measurement, or of the natural variability of SLA within the footprint of the radar measurement.

**Answer**: We agree. This paragraph has been largely rewritten. Now: "**Finally, this study does not aim to assess the accuracy of the SLA. It would only be possible by using co-located tide gauge observations as a reference. The results reflect the uncertainties in the estimated SLA related to errors in the processing and calculation algorithms. These uncertainties are quantified through the analysis of the STD of SLA obtained ssing different approaches in the calculation. In altimetry, this is a classical diagnosis of the algorithm performance, considering that a solution performs well when it reduces the variability in the SLA. As this study covers a wide range of algorithms, including the most recent and efficient algorithms available today to compute altimetry SLAs, it probably represents the best we can do today in estimating altimeter uncertainties.**"

Am I happy about the final sentence:

"Note that even if this work was carried out with LRM altimetry data, part of the conclusions should also contribute to modern altimetry techniques such as SAR and SARin, as all satellite

altimetry missions share some common correction terms, such as tidal and MSSH models for example. Even with their increased observational capabilities, which are favorable for monitoring coastal zones, the way these new types of altimetry observations are processed and the methodologies used to calculate the various geophysical corrections remain critical steps to derive accurate and precise geophysical information."

References

Missing References
- Laignel et al, 2022
- Andersen and Knudsen, 2009
- Pujol et al., 2016 (there is a Pujol et al., 2018)
- Schaeffer et al., 2022

References listed but not referenced in the text
- Egbert and Erofeeva, 2002
- Legeais et al., 2018
Peng and Deng, 2018 should be Peng et al., 2018

**Answer**: Thank you very much, the list has been corrected accordingly.

\*\*\*\*\*\*\*\*\*\*\*\*\*\*\*\*\*\*\*\*\*\*\*\*\*\*\*\*\*\*\*\*\*\*\*\*\*\*\*\*\*\*\*\*\*\*\*\*\*\*\*\*\*\*\*\*\*\*\*\*\*\*\*\*\*\*\*\*\*\*\*\*\*\*\*

**Answer to Reviewer 2**

This is a very useful work that help users of satellite altimetry data understand the limitations of the datasets in the open and coastal ocean domains. It also quantifies in a simple but precise and clear way the different possibilities that the user has to the apply geophysical corrections to the SLA. I have a few comments / questions that the authors might consider including before the article is being accepted in its final version.

**Answer**: Thank you, we're pleased that this work has been deemed useful and understandable for users. That was an important objective of this study.

20 Hz data are used, and it is assumed that the results obtained should remain valid for the 1 Hz dataset (L94-95). This is not necessarily true. The 1 Hz data are less noisy than the 20 Hz and therefore it is expected that the metrics developed in this work will be different. 20 Hz data are certainly preferred to get closer to the coast.

**Answer**: Right. We have removed this sentence.

The title states "Understanding uncertainties…". I will suggest replace "understanding" by "assessing", as no discussion on the article is provided to understand the origin of the uncertainties.
It will be great to include a discussion on the origin of the differences observed between the different algorithms and corrections.

**Answer**: We decided to keep the term 'understanding' because, although we don't go into detail about the causes of error in the various algorithms used to calculate the SLA, we still try to identify which type of term/correction becomes a source of error as we get closer to the coast. We feel that this is an important step towards identifying where efforts need to be made in the future to make further progress on the quality of coastal altimetry data. It goes beyond assessment.
Concerning the discussion on the origin of the differences observed between the different algorithms and corrections, we have added a few details in the conclusion: "In terms of origin, uncertainties in ocean tide models and in mean sea surface height models significantly contribute to the coastal SLA uncertainty budget in some regions. **About tidal models, despite major progress, the spatial resolution remains inadequate to take account of the dynamics of the near coastal tide (Hart Davis et al., 2024). Concerning MSSH solutions, they are still poorly constrained near the coast due to the lack of SLA data to calculate them and their poorer quality (Pujol et al., 2018).** The altimeter range and the SSB appear to be large contributors to SLA uncertainties in the open ocean but within 10 km off the coastline, they become the limiting factor in the use of altimetry data. **This is due to the complexity of radar echoes near the coast, which makes them much more difficult to model**."

Minor comments:

L94-95 20 Hz vs 1 Hz. See comment above

**Answer**: This sentence has been removed.

Legend Fig 1: looks incomplete. What are the blue dots?

**Answer**: Right. Now: "**In blue, geographical domains and segments of altimetry tracks (blue dots) considered in the Round Robin study**. …"

L143: there are more recent tidal model comparisons

**Answer**: This reference has been completed by Lyard et al., 2021.

L171: it is not the same list.

**Answer**: Right, corrected (indentation issue)

175: "not too deep inside estuaries or sheltered by islands" can you precise or provide a number instead of say "too deep"?

**Answer**: It is not possible to provide a number as it really depends on the area, so we have reformulated as follows (new text in bold):
"Stations located at a distance shorter than 50 km from a Jason2/3 nominal track**, avoiding locations sheltered by islands or inside estuaries** so that the ocean dynamics signals captured by the in situ instrument and the satellite altimeter are as similar as possible."

L225: why not compute the difference between the two iono corrections directly, instead to correct the SLA and then compare the two products?

**Answer**: We are not sure we understand this comment. We want to estimate the impact of errors in this correction on the SLA estimate and not compare the 2 ionospheric solutions. We also need to compare the SLA uncertainty associated to this correction with the SLA uncertainty associated to the other SLA components. Therefore we need to compute the same diagnostics.

328-329: sentence looks incomplete. Are negligible the omitted tidal components from EOT20 model in coastal areas or not? Perhaps just delete "that" in line 329.

**Answer**: This sentence was not clear and has been slightly rewritten. Now: "Indeed, the tidal components omitted in EOT20 are secondary, non linear elements that generally have larger amplitudes (at the millimeter or centimeter level) in shallow waters than in the deep waters of the open ocean (sub-millimeter)."

401: Tran et al 2022, reference not listed

**Answer**: The correct reference is Tran et al., 2021. It has been corrected.

OS questions
    Does the paper address relevant scientific questions within the scope of OS?
Yes
    Does the paper present novel concepts, ideas, tools, or data?
Yes

Are substantial conclusions reached?
Yes
   Are the scientific methods and assumptions valid and clearly outlined?
Yes
   Are the results sufficient to support the interpretations and conclusions?
Yes
   Is the description of experiments and calculations sufficiently complete and precise to allow their reproduction by fellow scientists (traceability of results)?
Yes
   Do the authors give proper credit to related work and clearly indicate their own new/original contribution?
Yes
   Does the title clearly reflect the contents of the paper?
A suggestion has been made

**Answer**: Please see the answer above.

   Does the abstract provide a concise and complete summary?
Yes
   Is the overall presentation well structured and clear?
Yes
   Is the language fluent and precise?
Yes
   Are mathematical formulae, symbols, abbreviations, and units correctly defined and used?
Yes
   Should any parts of the paper (text, formulae, figures, tables) be clarified, reduced, combined, or eliminated?
Yes. See specific comment above

**Answer**: Please see our answers above.

   Are the number and quality of references appropriate?
Yes
   Is the amount and quality of supplementary material appropriate?
Yes. Available on-line
* * *
**Answer to comment from Marcello Passaro**

I have appreciated the effort of the scientists to provide an evaluation of the different elements that play a role in computing the sea level anomaly. This is surely not an easy task.

**Answer**: Thank you, this study is indeed a lot of work.

Nevertheless I would appreciate some integrations and clarifications:

1) The authors have a great pool of tide gauges to use, which indeed are part of the evaluation of the tide models. These tide gauges are nowadays available at high rate thanks to public datasets such as GESLA. I am very puzzled about why the evaluation of ranges and SSBs is only based on internal criteria (the standard deviation approaching the coast) and ignores the possibility of an external evaluation based on in-situ data. Indeed, all previous efforts aimed at validating coastal sea level have used tide gauges as the best source of comparison. I would strongly recommend including such a comparison here as well, especially since the authors have already conducted this experiment and the results do not produce exactly the same retracker ranking as the standard deviation exerciese (see their presentation OSTST 2022, publicly available and attached here)

**Answer**: The aim of this paper is to quantify the uncertainties in the estimation of coastal altimetry SLAs that are directly associated with the choice of algorithms and corrections, not to validate the algorithms or assess their performance in terms of accuracy. It is an excerpt of a round-robin exercise that contains a lot more diagnoses, including systematic comparisons to in situ observations as you mention. All the results of this round-robin study are publicly available on the AVISO website (https://www.aviso.altimetry.fr/en/data/products/sea-surface-height-products/global/altimetry-innovative-coastal-approach-product-alticap/roundrobin-reports.html) but are out of the scope of this paper. Here, we only present results based on tide gauges in the case of the tidal corrections, in order to illustrate the disparity that can be observed depending on the regions (micro-tidal vs macro-tidal) for this correction particularly.

2) While I understand that the focus of the paper is on "uncertainties" and not on the dataset, this is still a round robin. In my opinion, Figure 10 shall not be displayed without an accompanying plot showing the amount of data available. The authors write "Note that the statistics associated with MLE4 are not completely comparable to those of the other retracking algorithms below 10 km because the number of data available at the ouput of the MLE4 retracker drops by about 20%, whereas the number of data available at the output of ALES and Adaptive remains stable up to about ~4-5 km from the coast.". I strongly recommend to show these numbers here (also the comparison between ALES and Adaptive), since I question the meaningfulness of the statistics if the number of available data for each dataset with respect to the distance to the coast is not shown, at least in an Appendix.

**Answer**: This is a good point, thank you. We have added a new pannel on figure 10 displaying the number of points available for each retracking as a function of the distance to the coast.

3) Can the authors comment on the paper about Figure 10b, which shows that ALES has the

best STD of SLA between 2.5 Km and 10 Km, which is exactly the area where the improvement from LRM retracking compared to standard open ocean processing is needed?

**Answer**: Our objective in this paper is not to assess and comment the performance of each algorithm separately, depending on the ocean area considered. It is to quantify the level of discrepancies between the currently available algorithms and how it translates in terms of uncertainty in the estimated SLAs as we get closer to the coast.

---

## Author Response (AR2)

*Thank you for responding carefully and thoroughly to the reviewers' suggestions and the community comment. I am very happy to accept the paper, congratulations!*

*I spotted the following wordings that felt awkward - please consider whether these suggestions would help readability, before you upload your final version for publication.*

**Answer:** Thank you for your proofreading. We are very pleased that this study has been accepted for publication in Ocean Science!

*Title: remove data, so it reads '…measurement of coastal sea level: insights….'*

**Answer**: done

L19 is dispersion the correct word?

**Answer**: It has been replaced by "spread"

L21 comparisons with rather than comparisons to

**Answer**: done

L22 should it be 'choice of mean sea surface' or perhaps 'uncertainties in the mean sea surface'?

**Answer**: Now: "uncertainties in the mean sea surface"

L22 should it be 'sea level measurement uncertainties'?

**Answer:** As sea level is not measured directly by altimetry but is derived from the radar measurements, we have chosen to use the word estimate rather than measurement. Now: "to uncertainties in sea level estimates"

L367 I would say 'Australian region'

**Answer**: done

L417 should ALES be defined on first use?

**Answer**: ALES was defined just above line 412.

L443 I would say datapoints rather than data here

**Answer**: now: "SLA estimate"

L543 I would say 'Concerning tidal models' or 'Considering tidal models' or 'Regarding tidal models'

**Answer**: now "concerning tidal models"

Please note also the comments from the copy editors: Some your tables contain coloured cells or/and coloured values. Please note that this will not be possible in the final revised version of the paper due to HTML conversion of the paper. When revising the final version, you can use footnotes or italic/bold font.

**Answer**: done